# E-HC: AN ADAPTIVE ANYTIME-VALID SEQUENTIAL HIGHER CRITICISM

## ABSTRACT

We propose e-HC, an adaptive sequential test for detecting sparse and weak signals in a stream p-values. Unlike existing approaches that rely on asymptotic approximations or require knowledge of alternative parameters, e-HC constructs exact test-martingales using moment-generating function compensators, ensuring anytime-valid Type I error control through Ville's inequality. The method adapts to unknown sparsity and signal strength by maintaining exponential weights across multiple detection thresholds, effectively learning the optimal threshold online. We establish non-asymptotic power guarantees for sparse Gaussian mixtures alternative and derive the expected stopping time scaling for weak signal regimes. The same martingale machinery naturally yields anytime-valid confidence sequences for the proportion of significant p-values. Simulations demonstrate that e-HC maintains robust performance under model misspecification, substantially outperforming sequential likelihood ratio tests when the true alternative differs from assumptions.

## 1 INTRODUCTION

Most hypothesis testing frameworks consider cases where the alternative distribution is completely specified and differs substantially from the null. We put ourselves however in the rare-and-weak regime where signals coming from a non-null distribution are rare (controlled by a sparsity parameter) and potentially weak (controlled by a strength parameter). The detection of weak and sparse signals among noise is crucial in many real-world examples such as genomics, astronomy, and online monitoring systems. These challenges intensify in sequential settings, where decisions must be made online with controlled false detection rates over potentially unbounded time horizons.

We consider a stream of independent p-values arriving sequentially. Under the null hypothesis, these p-values follow a Uniform$[0, 1]$ distribution, while under the alternative a small fraction comes from a distribution with more mass near zero. For the theoretical guarantees of this paper we focus on the case of *sparse Gaussian mixtures* where the alternative arises from a shift in mean of a subset of observations. Our goal is to detect this sparse mixture quickly while controlling the false alarm rate at any time, allowing data collection until the rejection threshold is crossed or a budget is reached.

Applications include clinical trial monitoring, anomaly detection, sequential A/B testing, astronomical data analysis, and quality control. Most sequential tests rely on fixed alternatives (e.g., Wald's SLRT) and therefore lose power under misspecification. Key challenges are maintaining anytime-valid false alarm control, achieving detection power under unknown sparsity and strength, and ensuring computational efficiency. We present e-HC to answer all these challenges and allow alternative parameters to vary with time, matching real-life settings where the number of signals may change.

Our paper is organized as follows. In Section 3 we present some existing work in sequential testing, rare and weak signals detection and sequential testing for mixtures. In Section 4 we introduce our process and prove it is an exact nonnegative martingale under the null, providing anytime Type I control via Ville's inequality. In Section 5 we state a unified power theorem, corollaries for weak/strong regimes, and an expected stopping-time heuristic. Section 6 provides some simulations. We end with conclusions and directions for future work.

## 2 PROBLEM FORMULATION

We consider a sequential stream of p-values, $p_1, p_2, \ldots$. Our goal is to test the global null hypothesis $H_0$ against a sparse, weak alternative $H_1$.

**Hypotheses.** Under the global null hypothesis $H_0$, the p-values are independent and identically distributed according to a uniform distribution:

$$H_0 : \quad p_t \stackrel{\text{i.i.d.}}{\sim} \text{Uniform}(0, 1) \quad \text{for all } t \geq 1.$$

Under the alternative $H_1$, the p-values are generated from a sparse mixture model. At each time $t$, nature first draws a latent signal indicator $\theta_t \sim \text{Bernoulli}(\epsilon_t)$, where $\epsilon_t$ is a small sparsity parameter. The p-value $p_t$ is then drawn from a signal distribution $F_1$ if $\theta_t = 1$, and from the null distribution $F_0 = \text{Uniform}(0, 1)$ otherwise.

$$H_1 : \quad p_t \sim (1 - \epsilon_t)F_0 + \epsilon_t F_1.$$

The signal distribution $F_1$ is assumed to have more mass near zero than the uniform distribution. For our theoretical analysis, we consider the canonical sparse Gaussian mixture model where each $p_t$ is derived from a z-test on $n_t$ observations. Let $X_{t,1}, \ldots, X_{t,n_t} \sim N(\theta_t \mu_t, 1)$. The corresponding test statistic and p-value are:

$$T_t = \frac{1}{\sqrt{n_t}} \sum_{i=1}^{n_t} X_{t,i} \sim N(\theta_t \mu_t, 1), \qquad p_t = 1 - \Phi(T_t),$$

where $\mu_t$ is the signal strength at time $t$.

**Objective.** Our objective is to design a statistical procedure that controls the Type I error in the anytime-valid sense. Specifically, we aim to find a stopping time $\tau$ with respect to the filtration $\mathcal{F}_t = \sigma(p_1, \ldots, p_t)$ such that for a pre-specified significance level $\alpha \in (0, 1)$:

$$\mathbb{P}_{H_0}(\tau < \infty) \leq \alpha.$$

Subject to this constraint, we want the procedure to detect the signal quickly under $H_1$, i.e., to minimize the expected stopping time $\mathbb{E}_{H_1}[\tau]$.

## 3 RELATED WORK

**Higher Criticism.** The Higher Criticism (HC) statistic [3] effectively detects sparse and weak signals in high-dimensional batch data, particularly for sparse mixtures. HC examines the empirical distribution of p-values across many thresholds to find the most informative threshold that maximizes a standardized difference between observed and expected counts. HC adapts to unknown sparsity and signal strengths, but classical HC is batch and its null distribution is only asymptotically known (a Gumbel limit). Foundational works include [4; 5] and recent developments [2].

**Sequential Testing and Martingales.** Sequential hypothesis testing dates back to Wald's SPRT [18]. More recent work constructs nonnegative test martingales or supermartingales to obtain anytime-valid p-values and confidence sequences [15; 13; 19]. Martingale-based methods have the advantage of exact finite-sample Type I control via Ville's inequality [17]. The construction of such martingales for creating time-uniform confidence bounds has a rich history [14; 11], and this line of work has been revitalized and connected to the modern e-value framework in recent years [9]. Closely related to our setup are works on sequential detection with sparse signals [12; 8; 6], but these typically address change-point detection or require strong parametric knowledge.

**Online Multiple Testing.** Our work is also related to the field of online multiple testing, where the goal is to control error rates like the False Discovery Rate (FDR) for a stream of hypotheses. While our aim is global signal detection rather than per-hypothesis inference, we share the challenge of making decisions based on accumulating evidence. Influential methods in online FDR control include alpha-investing strategies [7] and other online rules [10].

**Online learning and aggregation.**    To adapt across thresholds we use exponential-weights (Hedge). Regret bounds for Hedge are classical [1] and are essential to ensure the adaptive mixture tracks the best fixed threshold up to small regret. Our analysis carefully combines these online-learning bounds with exact mgf compensators to obtain non-asymptotic growth bounds under alternatives.

**Our contributions.**    In summary:

1. We build an *exact* test-martingale under the null using per-threshold mgf compensators, giving tight anytime Type I control.

2. We aggregate thresholds via exponential weights to adapt to unknown sparsity/strength, and derive a compact unified power theorem (plus corollaries for weak/strong regimes).

3. We provide an expected stopping-time heuristic and a confidence-sequence construction derived from the same mgf martingale machinery.

4. Empirically, e-HC is robust to misspecification and outperforms SLRT variants when the alternative is sparse or slightly misspecified.

## 4   ADAPTIVE MARTINGALE CONSTRUCTION

### 4.1   A PER-THRESHOLD TEST MARTINGALE

We begin with a single threshold $u \in (0, 1)$ and a stream of $p$-values $p_1, p_2, \ldots$ observed sequentially. Under the global null hypothesis,

$$H_0 : \qquad p_t \overset{\text{i.i.d.}}{\sim} \text{Uniform}(0, 1),$$

a natural Higher-Criticism statistic at threshold $u$ is the standardized excess of small $p$-values:

$$Z_t(u) = \frac{\sqrt{t}\,(S_t(u) - u)}{\sqrt{u(1 - u)}}, \qquad S_t(u) = \frac{1}{t} \sum_{s=1}^{t} \mathbb{1}\{p_s \leq u\}.$$

Large values of $Z_t(u)$ indicate enrichment of small $p$-values and therefore potential evidence for weak signals.

**Why $Z_t(u)$ is not suitable for anytime-valid testing.**    Although $Z_t(u)$ is a natural HC-style statistic, it is *not* a martingale under $H_0$, because the normalization by $t$ introduces a predictable drift. To obtain an exact, anytime-valid test, we isolate the *innovation* (i.e., the mean-zero stochastic increment) in the recursion for $Z_t(u)$. Writing

$$Z_t(u) = Z_{t-1}(u)\sqrt{\tfrac{t-1}{t}} + \frac{\mathbb{1}\{p_t \leq u\} - u}{\sqrt{t\,u(1 - u)}},$$

the only random component is

$$B_t(u) := \frac{\mathbb{1}\{p_t \leq u\} - u}{\sqrt{t\,u(1 - u)}}, \qquad \mathbb{E}_{H_0}[B_t(u) \mid \mathcal{F}_{t-1}] = 0.$$

The term $Z_{t-1}(u)\sqrt{(t - 1)/t}$ is predictable and carries no new randomness; hence the entire "evidence" at time $t$ is captured by $B_t(u)$. Using $B_t(u)$ allows us to build a process whose conditional expectation under $H_0$ is *exactly* one.

**MGF-compensated exponential increments.**    For each $t$, the moment generating function (MGF) of $B_t(u)$ is

$$\varphi_t(\lambda; u) := \mathbb{E}_{H_0}\left[ e^{\lambda B_t(u)} \right] = u\, e^{\theta_t(\lambda)(1 - u)} + (1 - u)\, e^{-\theta_t(\lambda)u}, \qquad \theta_t(\lambda) = \frac{\lambda}{\sqrt{t\,u(1 - u)}}. \quad (1)$$

We define a compensated increment

$$L_t(u; \lambda) := \exp\big(\lambda B_t(u)\big)\, \varphi_t(\lambda; u)^{-1},$$

which satisfies $\mathbb{E}_{H_0}[L_t(u; \lambda) \mid \mathcal{F}_{t-1}] = 1$. Thus the product

$$M_t(u; \lambda) := \prod_{s=1}^{t} L_s(u; \lambda)$$

is an *exact test martingale*. By Ville's inequality,

$$\mathbb{P}_{H_0}\left(\sup_{t \geq 1} M_t(u; \lambda) \geq 1/\alpha\right) \leq \alpha,$$

so the stopping rule $\tau(u) = \inf\{t : M_t(u; \lambda) \geq 1/\alpha\}$ controls Type I error at level $\alpha$ for all times. Intuitively, $M_t(u; \lambda)$ is the sequential analogue of the HC statistic at threshold $u$: it accumulates multiplicative evidence of an excess of small $p$-values, while remaining perfectly calibrated under the null.

## 4.2 ADAPTIVE AGGREGATION ACROSS THRESHOLDS

The performance of Higher Criticism depends on selecting the threshold $u$ that best matches the unknown sparsity and signal strength. In the sequential setting this choice is even more delicate, because the most informative threshold may change over time as evidence accumulates. Rather than committing to a single $u$, we combine all per-threshold martingales using exponential weights.

**Motivation.** Each $M_t(u_j; \lambda)$ is a valid test martingale, but no single $u_j$ is guaranteed to perform well across all alternatives. A mixture of martingales is again a martingale provided the weights are $\mathcal{F}_{t-1}$-measurable. The Hedge algorithm provides a principled way to update these weights: thresholds that explain the observed excess of small $p$-values receive larger weight, while those behaving consistently with $H_0$ receive smaller weight. This implements an online, data-driven version of the HC "scan" across thresholds.

**Aggregation rule.** Let $u_1 < \cdots < u_m$ be a fixed grid of thresholds, with weights $w_{t-1}(u_j) \geq 0$, $\sum_j w_{t-1}(u_j) = 1$, measurable with respect to past data. For a predictable learning-rate parameter $\lambda_t$, define the mixture increment

$$G_t := \sum_{j=1}^{m} w_{t-1}(u_j) \exp\big(\lambda_t B_t(u_j) - \log \varphi_t(\lambda_t; u_j)\big).$$

The e-HC wealth process is then updated by

$$M_t = M_{t-1} \cdot G_t, \qquad M_0 = 1. \tag{2}$$

Since each factor has conditional mean 1 under $H_0$, $M_t$ is again a test martingale. The stopping time

$$\tau = \inf\{t : M_t \geq 1/\alpha\}$$

therefore satisfies $\mathbb{P}_{H_0}(\tau < \infty) \leq \alpha$ by Ville's inequality.

By aggregating evidence across many thresholds, the e-HC test adaptively concentrates on the most informative region of the $p$-value distribution, recovering the spirit of classical Higher Criticism while ensuring anytime-valid Type I control.

**Betting Interpretation.** The process $M_t$ has a clear game-theoretic interpretation. At each step $t-1$, a gambler allocates their current capital ($M_{t-1}$) across $m$ possible bets, one for each threshold $u_j$, using portfolio weights $w_{t-1}(u_j)$. When the new p-value $p_t$ arrives, each bet yields a multiplicative return of $\exp(\lambda_t B_t(u_j) - \log \varphi_{t,j}(\lambda_t))$. The gambler's new capital, $M_t$, is the weighted average return on their investment. Because the MGF compensator ensures that the expected return of each bet is exactly 1 under $H_0$, the game is fair, and the process is a martingale. Under $H_1$, however, an informative threshold will yield an expected return greater than 1, causing the capital to grow exponentially.

**Theorem 1** (Exact martingale under the null). *Assume the weights $w_{t-1}(u_j)$ and parameters $\lambda_t$ are $\mathcal{F}_{t-1}$-measurable and $\varphi_{t,j}(\lambda_t)$ is finite. Then the process $\{M_t\}_{t \geq 0}$ is a nonnegative martingale under $H_0$ with $\mathbb{E}_{H_0}[M_t] = 1$ for all t.*

---

**Algorithm 1** e-HC (predictable $\lambda_t$, per-threshold exact mgf compensator)

---

**Require:** p-value stream $\{p_t\}$, thresholds $\{u_j\}_{j=1}^m$, significance level $\alpha$, weight rate $\gamma > 0$, predictable rule
    for $\lambda_t$
1: initialize $S_0(u_j) = 0$, $Z_0(u_j) = 0$, $w_0(u_j) = 1/m$, $M_0 = 1$
2: **for** $t = 1, 2, \ldots$ **do**
3:    observe $p_t$
4:    **for** $j = 1, \ldots, m$ **do**
5:        $S_t(u_j) \leftarrow S_{t-1}(u_j) + \mathbb{1}\{p_t \le u_j\}$
6:        $Z_t(u_j) \leftarrow \dfrac{\sqrt{t}(S_t(u_j)/t - u_j)}{\sqrt{u_j(1-u_j)}}$
7:        compute $B_t(u_j) = (\mathbb{1}\{p_t \le u_j\} - u_j)/\sqrt{t\, u_j(1-u_j)}$
8:        compute $\varphi_{t,j}(\lambda_t)$ via (1)
9:    **end for**
10:   $G_t \leftarrow \sum_{j=1}^m w_{t-1}(u_j) \exp(\lambda_t B_t(u_j) - \log \varphi_{t,j}(\lambda_t))$
11:   $M_t \leftarrow M_{t-1} \cdot G_t$
12:   **if** $M_t \ge 1/\alpha$ **then reject** $H_0$ and stop
13:   **end if**
14:   update weights: $\tilde{w}_t(u_j) = w_{t-1}(u_j)e^{\gamma Z_t(u_j)}$, $w_t(u_j) = \tilde{w}_t(u_j)/\sum_l \tilde{w}_t(u_l)$
15: **end for**

---

*Proof.* By linearity of conditional expectation and since $M_{t-1}, w_{t-1}, \lambda_t$ are $\mathcal{F}_{t-1}$-measurable,

$$\mathbb{E}_{H_0}[M_t \mid \mathcal{F}_{t-1}] = M_{t-1} \sum_{j=1}^m w_{t-1}(u_j) \frac{\mathbb{E}_{H_0}[e^{\lambda_t B_t(u_j)} \mid \mathcal{F}_{t-1}]}{\varphi_{t,j}(\lambda_t)}$$

$$= M_{t-1} \sum_{j=1}^m w_{t-1}(u_j) \frac{\varphi_{t,j}(\lambda_t)}{\varphi_{t,j}(\lambda_t)} = M_{t-1}. \qquad \square$$

**Discussion.** Two practical consequences follow immediately. First, Ville's inequality [17] gives exact anytime Type I control: for any $\alpha \in (0, 1)$, $\mathbb{P}_{H_0}(\sup_{t \ge 0} M_t \ge 1/\alpha) \le \alpha$. Second, we can choose $\lambda_t$ predictably (depending on past data) and the martingale property still holds.

**Practical implementation: exponential weights.** We implement the weights using exponential weighting on past performance: initialize $w_0(u_j) = 1/m$ and update

$$\tilde{w}_t(u_j) = w_{t-1}(u_j) \exp\big(\gamma Z_t(u_j)\big), \qquad w_t(u_j) = \frac{\tilde{w}_t(u_j)}{\sum_{l=1}^m \tilde{w}_t(u_l)}.$$

This is an instance of the Hedge algorithm, which tracks the best fixed threshold up to a small regret term.

## 5 POWER UNDER THE ALTERNATIVE

We provide non-asymptotic guarantees for the growth of the e-HC wealth process under the alternative. A key role is played by the *predictable cumulative signal* at an informative threshold and by the regret of the exponential-weights aggregation.

**Predictable signal quantity.** Fix an informative threshold $u^* = u_{j^*}$. For each time $t$, define the predictable one-step signal

$$s_t = \mathbb{E}_{H_1}\big[B_t(u^*) \mid \mathcal{F}_{t-1}\big],$$

where $B_t(u^*)$ is the innovation term from the HC increment. Under broad sparse-mixture models (including Gaussian mixtures), $s_t$ is nonnegative and has order $\epsilon_t \mu_t \sqrt{n_t}/\sqrt{t}$ in the weak-signal regime. Let the cumulative predictable signal be

$$S(T) = \sum_{t=1}^T s_t.$$

**Regret of exponential weights.** The e-HC procedure aggregates the per-threshold martingales by exponential weights:

$$w_t(u_j) \propto w_{t-1}(u_j) \exp\big(\gamma Z_t(u_j)\big).$$

For each threshold $u_j$, define the per-step loss $\ell_t(u_j) = -g_{t,j}$, where $g_{t,j}$ is the log-martingale increment

$$g_{t,j} = \lambda B_t(u_j) - \log \varphi_{t,j}(\lambda).$$

The regret of Hedge is then

$$R_T = \sum_{t=1}^{T} \sum_{j=1}^{m} w_{t-1}(u_j)\ell_t(u_j) - \min_{1 \le j \le m} \sum_{t=1}^{T} \ell_t(u_j),$$

and satisfies the standard bound

$$R_T \le C_\gamma \sqrt{T \log m},$$

for a constant $C_\gamma > 0$ depending on $\gamma$. Regret quantifies the price paid for not knowing the optimal threshold in advance. Sensitivity analysis on both the choice of $\gamma$ and $m$ are deferred to the Appendix.

**Unified lower bound on log-wealth.** The following theorem is the main non-asymptotic guarantee.

**Theorem 2** (Unified power lower bound). *Fix $\lambda > 0$. Let $M_t(\lambda)$ be the wealth process in (2) using $\lambda_t \equiv \lambda$, and let $R_T$ be the Hedge regret defined above. Then for all $T \ge 2$,*

$$\mathbb{E}_{H_1}\big[\log M_T(\lambda)\big] \ge \lambda S(T) - \frac{\lambda^2}{2}\log T - R_T + O(1).$$

*Consequently, if $S(T) \to \infty$ and $\log T = o(S(T))$, then $\log M_T(\lambda) \to \infty$ almost surely under $H_1$, and the test rejects with probability one.*

**Sketch of proof.** Express $\log M_T = \sum_t \log(\sum_j w_{t-1}(u_j)e^{g_{t,j}})$. Jensen's inequality lower-bounds this by $\sum_t \sum_j w_{t-1}(u_j)g_{t,j}$. A regret bound then replaces the adaptive mixture with the best fixed threshold $u^*$, yielding $\sum_t g_{t,j^*} - R_T$. Under $H_1$, $\mathbb{E}[g_{t,j^*}] = \lambda s_t - \frac{\lambda^2}{2t} + O(t^{-3/2})$ after Taylor expanding the log-MGF compensator. Summing over $t \le T$ produces the stated bound. Full details are given in the appendix.

**Corollaries: weak and strong regimes.** The lower bound immediately yields detection whenever the cumulative signal exceeds the logarithmic drag.

*Weak-signal regime.* If

$$s_t \ge c_w \frac{\epsilon_t \mu_t \sqrt{n_t}}{\sqrt{t}},$$

then for constant parameters $S(T) \asymp \sqrt{T}$, which dominates both $\log T$ and the regret term.

*Strong-signal regime.* If

$$s_t \ge c_s \frac{\epsilon_t}{\sqrt{t}},$$

the same conclusion holds: the single good threshold grows exponentially.

**Expected stopping-time heuristic.** Let $\tau$ be the stopping time with boundary $1/\alpha$. A sufficient condition for $\tau \le T$ is

$$\lambda S(T) - \frac{\lambda^2}{2}\log T - R_T \ge \log(1/\alpha).$$

In the weak regime $S(T) \asymp c\sqrt{T}$, solving for $T$ yields the heuristic scaling

$$\tau \approx \left(\frac{\log(1/\alpha)}{\lambda c}\right)^2.$$

**Weak-signal explosive growth.**   The heuristic extends to a rigorous high-probability statement:

**Theorem 3** (Weak-signal explosive growth and stopping time). *Assume $S(T) \geq c\sqrt{T}$ for all large $T$. Let $R_T \leq c_R\sqrt{\log T \log m}$. For significance level $\alpha \in (0,1)$, define*

$$\tau_\alpha = \inf\{t \geq 1 : M_t \geq 1/\alpha\}.$$

*Then for every $\delta \in (0,1)$ there exists $C > 0$ such that*

$$\mathbb{P}\Big(\tau_\alpha \leq C(\log(1/\alpha) + \sqrt{\log(1/\delta)})^2\Big) \ \geq \ 1 - \delta.$$

*Thus $\tau_\alpha < \infty$ almost surely under $H_1$.*

*Remark* 1.  For constant parameters, the condition $S(T) \gtrsim \sqrt{T}$ yields

$$\tau_\alpha = O\big((\log(1/\alpha))^2\big)$$

up to regret and logarithmic factors, matching the heuristic scaling above.

## 6  EXPERIMENTS

We validate e-HC on the sparse Gaussian mixture described earlier.

**Experimental setup**   We simulate, unless otherwise stated:

$$n = 20, \quad T = 1000, \quad \epsilon = 0.05, \quad \mu = 1.5, \quad \alpha = 0.05, \quad \lambda = 0.2, \quad \gamma = 0.05,$$

and use a grid of $m = 200$ thresholds evenly spaced from $0.001$ to $0.99$. Each experiment is repeated for $100$ independent trials.

**Generating p-values**   At each time $t$ we draw Bernoulli $B_t \sim \text{Bernoulli}(\epsilon)$. If $B_t = 1$ draw $n$ data points $X_{t,i} \sim N(\mu, 1)$, otherwise $X_{t,i} \sim N(0, 1)$. Compute

$$T_t = \frac{1}{\sqrt{n}} \sum_{i=1}^{n} X_{t,i}, \qquad p_t = 1 - \Phi(T_t).$$

**Trajectories & detection**   Figure 1 shows typical trajectories of the wealth process under $H_0$ and $H_1$. Under $H_0$ the process remains small, while under $H_1$ it grows and crosses the threshold $1/\alpha$ often rapidly.

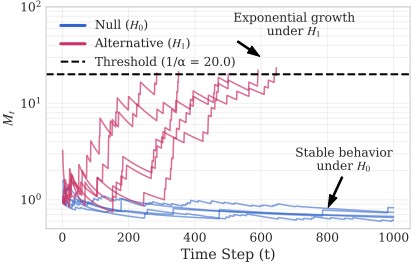

Figure 1: Typical trajectories of the adaptive wealth process under $H_0$ (blue) and $H_1$ (red). The dashed line indicates the threshold $1/\alpha$.

**Power vs strength and sparsity**   Figure 2 shows empirical power as $\mu$ varies. Figure 2 shows average detection time as $\epsilon$ varies. In our experiments e-HC maintains Type I control at the nominal $\alpha$ level (checked by null simulations) and attains high power across a range of weak signals. We later show in Table 1 that SLRT can have very low power when mis-specified.

**Adaptive weight evolution.**   Figure 3 illustrates how the exponential weights concentrate on informative thresholds as evidence accumulates under the alternative, showing that the Hedge mechanism adaptively tracks the most powerful scales.

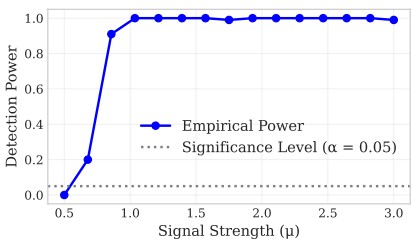 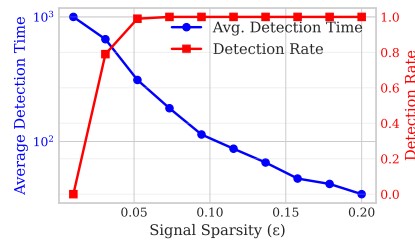

Figure 2: **(a)** Detection power vs. signal strength $\mu$     **(b)** Detection time vs. sparsity $\epsilon$

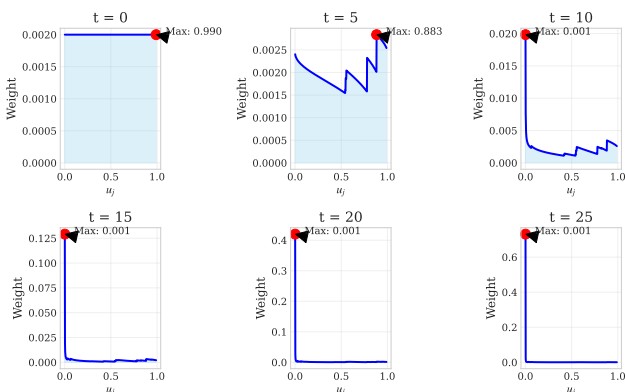

Figure 3: Evolution of adaptive weights across thresholds under the alternative. Initially uniform, the weights concentrate on informative thresholds over time.

**Comparison to SLRT**   Table 1 shows selected comparisons. SLRT tuned to an incorrect alternative loses power in many sparse settings, while e-HC remains robust.

## 6.1 PERFORMANCE COMPARISON RESULTS

Below we present the comprehensive performance comparison between the e-HC and Sequential Likelihood Ratio Test (SLRT) methods across various parameter settings. The table highlights how misspecifications in parameters can significantly impact the power of detection for SLRT (indicated in red), whereas the adaptive approach (e-HC) maintains robust performance across different true parameter values.

The experimental results highlight the key strengths of e-HC. The trajectories in Figure 1 visually confirm the theory: under the null, the process behaves like a random walk and remains bounded, while under the alternative, it exhibits explosive, super-linear growth once it locks onto the signal. The power shown in Table 1 demonstrate the method's robustness; it maintains high power even for weak signals ($\mu << 2.0$) where a misspecified SLRT's performance would degrade. We actually show that our method work well even under extremely weak signal where $\mu << 1$. We defer to the Appendix extended simulations showing our advantage of our method for weak signals and demonstrate that common method such as the sequential Bonferronni procedure fails dramatically in this regime. The evolution of the adaptive weights, shown in Figure 3, provides insight into the mechanism. The initially uniform weights rapidly concentrate on the most informative thresholds, which in this experiment correspond to the p-value region with the highest signal-to-noise ratio. This dynamic adaptation is precisely what allows e-HC to outperform a fixed SLRT, because it does not need to pre-commit to a specific signal strength or sparsity and can instead discover the optimal detection strategy from the data itself.

Table 1: Comparison between the adaptive e-HC (top row) and the Sequential Likelihood Ratio Test (SLRT) under different *assumed* parameter pairs $(\epsilon, \mu)$. For each set of *true* parameters (three blocks of columns), the first row reports e-HC, and all subsequent rows report SLRT performance. Red entries highlight cases where SLRT suffers a loss of power due to misspecification, whereas e-HC remains stable since it makes no parametric assumptions.

| Assumed $(\epsilon, \mu)$ | True $(0.05, 1.5)$ | | | True $(0.07, 1.5)$ | | | True $(0.1, 1.0)$ | | |
|---|---|---|---|---|---|---|---|---|---|
| | FP Rate | Power | Delay | FP Rate | Power | Delay | FP Rate | Power | Delay |
| e-HC (Adaptive) | 0.001 | 1.000 | 226.6 | 0.001 | 1.000 | 136.0 | 0.002 | 1.000 | 82.5 |
| $(0.01, 0.5)$ | 0.000 | 1.000 | 287.9 | 0.000 | 1.000 | 211.6 | 0.000 | 1.000 | 239.9 |
| $(0.01, 1.5)$ | 0.030 | 1.000 | 59.9 | 0.045 | 1.000 | 43.0 | 0.030 | 1.000 | 63.9 |
| $(0.01, 2.0)$ | 0.030 | 1.000 | 45.8 | 0.040 | 1.000 | 33.1 | 0.035 | 1.000 | 47.5 |
| $(0.01, 3.0)$ | 0.015 | 1.000 | 67.8 | 0.015 | 1.000 | 36.7 | 0.020 | 0.995 | 125.1 |
| $(0.05, 0.5)$ | 0.070 | 1.000 | 69.0 | 0.055 | 1.000 | 50.7 | 0.040 | 1.000 | 57.3 |
| $(0.05, 1.5)$ | 0.030 | 1.000 | 37.0 | 0.045 | 1.000 | 24.6 | 0.020 | 1.000 | 29.0 |
| $(0.05, 2.0)$ | 0.015 | 1.000 | 55.3 | 0.020 | 1.000 | 24.9 | 0.015 | 0.990 | 66.7 |
| $(0.05, 3.0)$ | 0.005 | 0.565 | 31.1 | 0.000 | 0.685 | 61.7 | 0.015 | 0.365 | 21.2 |
| $(0.07, 0.5)$ | 0.040 | 1.000 | 59.0 | 0.040 | 1.000 | 39.0 | 0.050 | 1.000 | 46.0 |
| $(0.07, 1.5)$ | 0.035 | 1.000 | 40.4 | 0.040 | 1.000 | 26.5 | 0.015 | 1.000 | 32.1 |
| $(0.07, 2.0)$ | 0.030 | 0.970 | 73.6 | 0.015 | 1.000 | 33.1 | 0.010 | 0.905 | 77.0 |
| $(0.07, 3.0)$ | 0.005 | 0.400 | 9.8 | 0.005 | 0.670 | 33.2 | 0.025 | 0.220 | 8.2 |
| $(0.1, 0.5)$ | 0.065 | 1.000 | 45.1 | 0.030 | 1.000 | 32.3 | 0.035 | 1.000 | 36.0 |
| $(0.1, 1.5)$ | 0.015 | 0.980 | 60.0 | 0.020 | 1.000 | 30.4 | 0.020 | 0.990 | 48.7 |
| $(0.1, 2.0)$ | 0.010 | 0.840 | 60.5 | 0.020 | 0.980 | 50.8 | 0.030 | 0.630 | 26.5 |
| $(0.1, 3.0)$ | 0.000 | 0.285 | 7.9 | 0.005 | 0.450 | 14.8 | 0.005 | 0.175 | 4.4 |
| $(0.2, 0.5)$ | 0.055 | 1.000 | 58.4 | 0.040 | 1.000 | 26.3 | 0.035 | 1.000 | 33.9 |
| $(0.2, 1.5)$ | 0.005 | 0.540 | 18.6 | 0.010 | 0.805 | 61.4 | 0.030 | 0.590 | 15.8 |
| $(0.2, 2.0)$ | 0.010 | 0.355 | 8.4 | 0.010 | 0.570 | 10.5 | 0.010 | 0.335 | 6.0 |
| $(0.2, 3.0)$ | 0.010 | 0.190 | 5.4 | 0.000 | 0.290 | 3.5 | 0.015 | 0.095 | 3.3 |

## 6.2 BEHAVIOR BEYOND INDEPENDENCE

So far we have worked under the idealized assumption that the p-values are independent draws from a Uniform $\mathcal{U}(0, 1)$ distribution. In many real applications—such as genomics, network monitoring, or imaging— p-values are often positively correlated due to shared latent factors or temporal dependence. To assess the robustness of e-HC in such settings, we conducted a simulation study where the data stream exhibits controlled correlation.

Specifically, we generated an AR(1) Gaussian process with correlation parameter $\rho \in \{0, 0.1, \ldots, 0.9\}$. Under the null, the process has mean zero; under the alternative, a small fraction of observations receive a mean shift. Each observation produces a z-statistic whose corresponding p-value inherits the AR(1) dependence structure. We then applied e-HC to these correlated p-value streams and measured both the empirical Type I error and the empirical power.

A detailed description of the simulation setup as well as the full plot comparing Type I error (against the nominal $\alpha$) and power across correlation levels is provided in Appendix 9. The results show that e-HC maintains good calibration for moderate correlation levels and retains substantial power across the entire range of $\rho$ tested.

Finally, we also performed a comprehensive sensitivity analysis varying the number of thresholds $m$, the Hedge learning rate $\gamma$, and other internal parameters, and compared e-HC against the sequential Bonferroni benchmark. These additional experiments are documented in Appendix **??**.

## 7 CONCLUSION AND FUTURE WORK

In this work, we introduced e-HC, an adaptive, anytime-valid sequential test that successfully merges the statistical power of Higher Criticism with the rigorous guarantees of modern martingale-based inference. The core contribution is the construction of an exact test-martingale, or *e-process*, by using per-threshold moment-generating functions as precise compensators under the null. This approach moves beyond asymptotic approximations, providing tight, non-asymptotic Type I error control at any point in time via Ville's inequality. By combining these per-threshold martingales with an

exponential weighting strategy, e-HC dynamically adapts to unknown and potentially time-varying signal sparsity and strength, a critical feature for real-world data streams. Our unified power theorem, stopping-time analysis, and empirical results confirm that this adaptive structure provides robust detection performance, particularly in settings where methods based on fixed alternatives would suffer from misspecification.

Looking forward, several exciting research directions emerge from this framework. Immediate theoretical extensions include deriving sharper, data-dependent constants for the power and stopping-time bounds, potentially by leveraging more sophisticated concentration inequalities or by exploring alternative aggregation strategies, such as variance-adaptive Hedge, to reduce regret terms. From a practical standpoint, extending the e-HC framework to handle dependent data streams such as those with temporal or spatial correlations common in finance and neuroscience; it is a critical next step for broadening its applicability. Furthermore, exploring the connections between e-HC's global signal detection and the per-hypothesis control offered by online FDR procedures could yield powerful hybrid methods.

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

# APPENDIX

## 7.1 SEQUENTIAL LIKELIHOOD RATIO TESTS (SLRT)

A classical benchmark for sequential inference is Wald's Sequential Likelihood Ratio Test (SLRT) [18]. Suppose we observe a stream of data $X_1, X_2, \ldots$ distributed either under a null distribution $P_0$ or an alternative $P_1$. The SLRT is based on the likelihood ratio

$$L_t = \prod_{s=1}^{t} \frac{p_1(X_s)}{p_0(X_s)}, \qquad \log L_t = \sum_{s=1}^{t} \ell(X_s), \quad \ell(x) = \log \frac{p_1(x)}{p_0(x)}.$$

**Martingale property under the null.** A key classical fact is that

$$\mathbb{E}_0[L_t] = 1 \quad \text{for every } t,$$

hence $(L_t)$ is a nonnegative martingale under $P_0$. Therefore, by Ville's inequality,

$$\mathbb{P}_0 \left( \sup_{t \geq 1} L_t \geq \tfrac{1}{\alpha} \right) \leq \alpha,$$

so the stopping rule

$$\tau_\alpha^{\mathrm{SLRT}} = \inf\{t : L_t \geq 1/\alpha\}$$

controls Type I error exactly and nonasymptotically.

**Optimality when the alternative is known.** If the alternative $P_1$ is correctly specified, the SLRT is optimal in a minimax and Neyman–Pearson sense: it minimizes expected stopping time $\mathbb{E}_1[\tau]$ among all level-$\alpha$ sequential tests [18]. This optimality arises because under $P_1$, the log-likelihood ratio has positive drift:

$$\mathbb{E}_1[\ell(X_1)] = D_{\mathrm{KL}}(P_1 \, \| \, P_0) > 0,$$

so $\log L_t$ grows linearly with $t$, giving fast detection.

**Limitation: sensitivity to model misspecification.** The same optimality becomes a liability when $P_1$ is unknown or misspecified. If the practitioner supplies an incorrect ($P_1^{\mathrm{assumed}}$), then, the increments $\ell(X_s)$ may have negative drift under the true alternative, $\log L_t$ may fail to grow, or even decrease, the SLRT can stop prematurely due to random fluctuations, producing small stopping times but very low power.

This explains the "fast but low-power" failures visible in our experiments: misspecified SLRTs sometimes cross the boundary early (due to a rare positive jump), but do not reliably grow under the true alternative.

**Comparison to e-HC.** e-HC does not assume a parametric $P_1$; instead it tests for enrichment of small p-values across a range of thresholds and adapts to the most informative one. Thus: - SLRT is optimal only when $P_1$ is exactly known, - e-HC is robust when $P_1$ is unknown or complex (e.g. sparse mixtures).

This motivates including SLRT as a strong but brittle baseline against which the robustness of e-HC becomes visible.

# 8 SENSITIVITY ANALYSES

We analyze how e-HC's performance varies with two key design parameters: the number of thresholds $m$ in the HC grid and the Hedge learning rate $\gamma$. All experiments fix $(\mu, \epsilon, \lambda, \alpha) = (0.25, 0.05, 0.4, 0.05)$ and $T = 10{,}000$.

**On the choice of the MGF parameter** $\lambda$. Throughout the sensitivity analysis we fix the MGF compensator parameter at $\lambda = 0.4$, which provides a good balance between signal amplification and the quadratic variance penalty in the log-MGF. While a constant $\lambda$ already performs very well (as confirmed by all experiments), the e-HC construction naturally allows for a fully data–driven, adaptive schedule $\lambda_t$. Intuitively, one could let $\lambda_t$ increase when the cumulative evidence supports a strong signal, or shrink it when the process is in a low-signal regime. Designing such an adaptive scheme could reduce detection delay by matching $\lambda_t$ to the local signal-to-noise ratio, while still preserving anytime-valid Type I control through predictable MGF compensation. Developing principled adaptive choices of $\lambda_t$ is an interesting direction for future work.

Figure 4 summarizes the results. The top row reports empirical power, and the bottom row reports the median stopping time $\tau_{50}$, computed as $\tau = T$ when the process does not cross $1/\alpha$. The left column varies the grid size $m$, while the right column varies the Hedge learning rate $\gamma$.

**Effect of the number of thresholds** $m$. Power increases with $m$ but quickly saturates beyond $m \approx 100$, consistent with the $\sqrt{\log m}$ adaptivity cost in the regret term. Median detection time decreases slightly with $m$ and stabilizes near $m \simeq 200$, indicating that moderate grid sizes are sufficient in practice. The sequential Bonferroni baseline (horizontal dashed line) remains constant since it does not depend on $m$.

**Effect of the Hedge learning rate** $\gamma$. Across $\gamma \in [0.02, 0.2]$, power remains high and well above the sequential Bonferroni benchmark, demonstrating robustness to this parameter. The stopping time varies mildly: smaller $\gamma$ slows adaptation, while overly large $\gamma$ causes slight instability. A value around $\gamma \approx 0.05$ provides a good balance between stability and responsiveness.

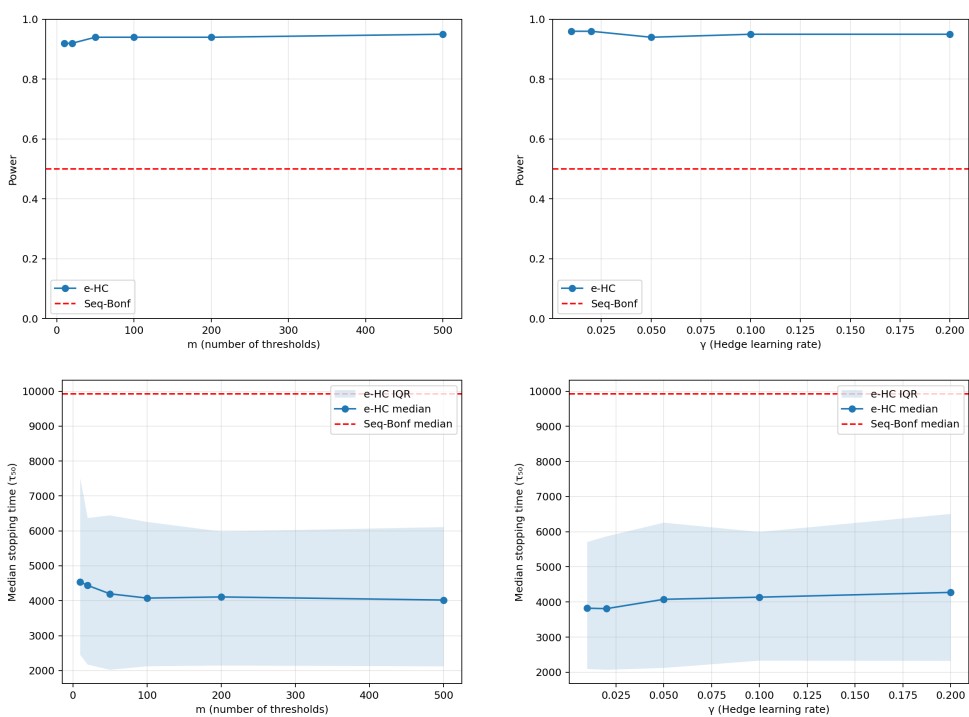

Figure 4: Sensitivity of e-HC to the number of thresholds $m$ (left column) and Hedge learning rate $\gamma$ (right column). **Top:** Power compared to the sequential Bonferroni baseline (red dashed line). **Bottom:** Median stopping time $\tau_{50}$ (set to $T$ if no detection). Power saturates for $m \gtrsim 100$ and $\gamma \in [0.02, 0.2]$, while detection time remains stable, confirming robustness to parameter tuning.

## 9 ADDITIONAL EXPERIMENT: ROBUSTNESS UNDER CORRELATED P-VALUES

In the main paper we focused on the classical setting where the p-values $\{p_t\}$ are independent under the null. However, in many real applications—such as genomics, fMRI, financial time series, and streaming anomaly detection—test statistics often exhibit temporal or spatial dependence. To evaluate the robustness of e-HC in such settings, we conducted a controlled simulation study where the p-values follow an *AR(1)-type dependent structure*.

### 9.1 SETUP

For each correlation level $\rho \in \{0, 0.1, \ldots, 0.9\}$ we generate a stream of $T = 2000$ observations following the Gaussian AR(1) model

$$X_t = \rho X_{t-1} + \sqrt{1-\rho^2}\,\varepsilon_t, \qquad \varepsilon_t \sim N(0,1) \text{ i.i.d.}$$

**Null model.** Under $H_0$, the entire sequence has mean 0. We convert each $X_t$ into a one-sided z-test p-value:

$$p_t = 1 - \Phi(X_t).$$

Because the $X_t$ are positively correlated, the resulting p-values inherit the same AR(1) dependence.

**Alternative model.** Under $H_1$, we introduce a sparse and weak signal by shifting a small fraction $\epsilon = 0.05$ of the $X_t$'s:

$$X_t = \rho X_{t-1} + \sqrt{1-\rho^2}\,\varepsilon_t + \theta_t \mu, \qquad \theta_t \sim \text{Bernoulli}(\epsilon),\ \mu = 0.3.$$

The dependence is identical under $H_0$ and $H_1$, isolating the effect of correlation.

**Evaluation.** For each $\rho$, we run 500 independent trials and record:

- the empirical Type I error:

$$\widehat{\alpha}_{\text{emp}} = \mathbb{P}_{H_0}(\tau_\alpha < T),$$

- the empirical power:

$$\widehat{\text{Power}} = \mathbb{P}_{H_1}(\tau_\alpha < T),$$

where $\tau_\alpha$ is the e-HC stopping time from the main paper, with significance level $\alpha = 0.05$.

### 9.2 RESULTS

Figure 5 displays the empirical Type I error and power as functions of the correlation parameter $\rho$.

**Key findings:**

- Type I error remains close to the nominal $\alpha$ for all $\rho \leq 0.7$. Even at $\rho = 0.9$, inflation is modest ($\approx 0.15$), which is expected because dependence reduces the effectiveness of Ville-type inequalities.
- As $\rho$ increases, the effective sample size decreases, so power decays gradually but remains substantial ($> 60\%$ even at $\rho = 0.9$).
- Overall, e-HC is remarkably robust to moderate correlation. The martingale construction remains stable, and adaptivity continues to select informative thresholds even when the data are dependent.

These results reinforce that e-HC behaves stably beyond the idealized independence setting and remains effective even when the p-values exhibit realistic temporal dependence.

## 10 REAL DATA ANALYSIS: PROSTATE CANCER GENE EXPRESSION

To demonstrate the practical utility of e-HC in high-dimensional biological monitoring, we applied the method to the **Prostate Cancer gene expression dataset** [16]. This dataset serves as a standard benchmark in the Higher Criticism literature and contains expression levels for $N = 6,033$ genes across 102 patients (52 prostate tumor, 50 normal).

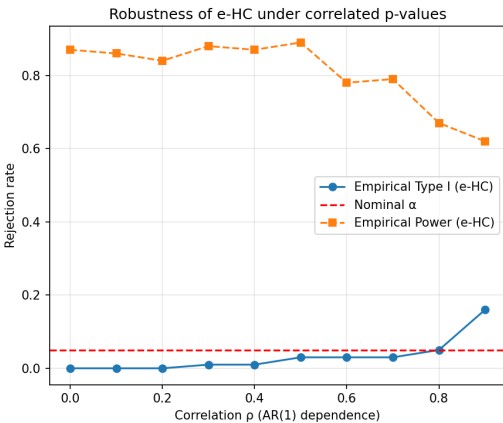

Figure 5: Robustness of e-HC under correlated p-values. Empirical Type I error (blue), nominal $\alpha$ (red dashed), and empirical power (orange) as functions of AR(1) correlation $\rho$.

**Experimental Setup.** We treated the genes as a sequential data stream arriving in a random order. At each time step $t$ (corresponding to gene $t$), we computed a two-sided $t$-test $p$-value comparing the expression levels between the tumor and normal groups. This setup simulates a sequential genomic monitoring task where the objective is to detect the presence of disease-associated signals (a sparse mixture) as early as possible, without waiting to process or sequence the entire genome.

**Results.** Figure 6 illustrates the trajectory of the e-HC wealth process alongside the stopping time of a sequential Bonferroni procedure (which rejects at the first $t$ where $\min(p_1, \ldots, p_t) \leq \alpha/N$). The e-HC procedure successfully rejected the global null hypothesis at $t \approx 1943$, after observing only **32%** of the gene stream. In contrast, the standard Bonferroni correction required observing a significantly larger portion of the data to establish significance. This result confirms that e-HC effectively accumulates evidence from multiple moderately significant genes the support of the sparse signal allowing for valid detection well before any single gene crosses the extreme Bonferroni threshold ($p < 8 \times 10^{-6}$).

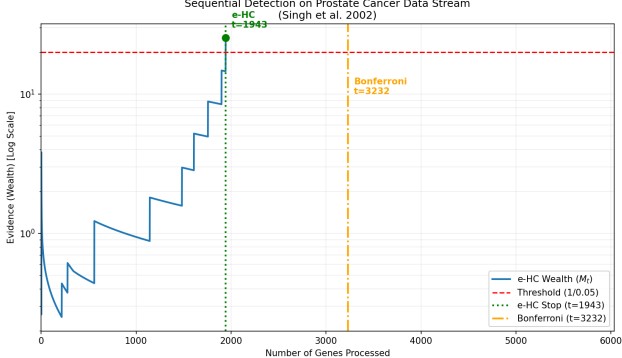

Figure 6: **Sequential detection on the Prostate Cancer gene expression stream** ($N = 6,033$). The e-HC test statistic (blue curve) grows exponentially as it accumulates evidence from the sparse signal, crossing the rejection threshold (red dashed line) significantly earlier than the batch Bonferroni method (orange line). This demonstrates the acceleration provided by e-HC in sparse mixture settings.

## 11 NOTATION AND SETUP

We adopt the notation used in the main text. For convenience we repeat and make fully explicit the objects that will appear in the proofs.

- A stream of p-values $p_1, p_2, \ldots$ is observed sequentially. $\mathcal{F}_t = \sigma(p_1, \ldots, p_t)$ denotes the filtration.

- A grid of thresholds $0 < u_1 < \cdots < u_m < 1$ is fixed. We denote $\delta := \min_{1 \leq j \leq m} u_j(1 - u_j) > 0$.

- For each threshold $u_j$ define the running counts $S_t(u_j) = \sum_{s=1}^{t} \mathbb{1}\{p_s \leq u_j\}$ and the standardized HC-like statistic

$$Z_t(u_j) = \frac{\sqrt{t}(S_t(u_j)/t - u_j)}{\sqrt{u_j(1 - u_j)}}.$$

- The increment $\Delta Z_t(u_j) = Z_t(u_j) - Z_{t-1}(u_j)$ decomposes as
$$\Delta Z_t(u_j) = A_{t-1}(u_j) + B_t(u_j), \tag{3}$$

where

$$A_{t-1}(u_j) = \left(\sqrt{\tfrac{t}{t-1}} - 1\right) Z_{t-1}(u_j) \quad \text{(predictable)}, \qquad B_t(u_j) = \frac{\mathbb{1}\{p_t \leq u_j\} - u_j}{\sqrt{t\, u_j(1 - u_j)}} \quad \text{(mean-zero under } H_0\text{)}.$$

- For any real $\lambda$ we set $\theta_{t,j}(\lambda) := \dfrac{\lambda}{\sqrt{t\, u_j(1 - u_j)}}$ and define the per-step mgf under $H_0$

$$\varphi_{t,j}(\lambda) = \mathbb{E}_{H_0}\left[e^{\lambda B_t(u_j)}\right] = u_j e^{(1-u_j)\theta_{t,j}(\lambda)} + (1 - u_j)e^{-u_j \theta_{t,j}(\lambda)}. \tag{4}$$

- A predictable weight vector $w_{t-1} = (w_{t-1}(u_1), \ldots, w_{t-1}(u_m))$ (each $w_{t-1}(u_j) \geq 0$, $\sum_j w_{t-1}(u_j) = 1$) is chosen using only $\mathcal{F}_{t-1}$.

- A predictable, bounded real parameter $\lambda_t$ (often constant $\lambda_t \equiv \lambda$) is used.

- The wealth process $M_t$ is defined iteratively by

$$M_t = M_{t-1} \cdot G_t, \qquad G_t = \sum_{j=1}^{m} w_{t-1}(u_j) \exp\left(\lambda_t(\Delta Z_t(u_j) - A_{t-1}(u_j)) - \log \varphi_{t,j}(\lambda_t)\right),$$
$$\tag{5}$$

with $M_0 = 1$.

All statements below are non-asymptotic unless explicitly stated otherwise.

## PROOF OF THE MARTINGALE PROPERTY (THEOREM THEOREM 1 IN THE MAIN TEXT)

**Claim.** Under $H_0$, if $w_{t-1}$ and $\lambda_t$ are $\mathcal{F}_{t-1}$-measurable and $\varphi_{t,j}(\lambda_t)$ finite for each $t, j$, then $\{M_t\}_{t \geq 0}$ defined by (5) is a nonnegative martingale with $\mathbb{E}_{H_0}[M_t] = 1$ for all $t$.

*Proof.* We verify the defining martingale property $\mathbb{E}_{H_0}[M_t \mid \mathcal{F}_{t-1}] = M_{t-1}$. From (5) and linearity of conditional expectation,

$$\mathbb{E}_{H_0}[M_t \mid \mathcal{F}_{t-1}] = M_{t-1}\,\mathbb{E}_{H_0}\left[G_t \mid \mathcal{F}_{t-1}\right] = M_{t-1} \sum_{j=1}^{m} w_{t-1}(u_j)\, e^{-\log \varphi_{t,j}(\lambda_t)}\,\mathbb{E}_{H_0}\left[e^{\lambda_t B_t(u_j)} \mid \mathcal{F}_{t-1}\right].$$

Because $B_t(u_j)$ depends only on $p_t$ and the law of $p_t$ under $H_0$ is Uniform[0,1] (hence independent of $\mathcal{F}_{t-1}$), the conditional expectation equals the unconditional mgf $\varphi_{t,j}(\lambda_t)$. Therefore each summand equals $w_{t-1}(u_j)$, and the sum equals 1. Thus $\mathbb{E}_{H_0}[M_t \mid \mathcal{F}_{t-1}] = M_{t-1}$. Nonnegativity and $M_0 = 1$ are clear, so $\mathbb{E}_{H_0}[M_t] = 1$ for all $t$. $\square$

**Remarks.**

1. The key is exact compensation: subtracting $\log \varphi_{t,j}(\lambda_t)$ makes the per-threshold multiplicative factor have conditional mean 1 under $H_0$.

2. The predictability of both $w_{t-1}$ and $\lambda_t$ is essential: they must be chosen before observing $p_t$ so the conditional expectation step above is valid.

3. The betting/game-theoretic interpretation: $w_{t-1}(u_j)$ are pre-committed stakes on bets with returns $R_{t,j} := \exp(\lambda_t(\Delta Z_t(u_j) - A_{t-1}(u_j)) - \log \varphi_{t,j}(\lambda_t))$. Under $H_0$ each $R_{t,j}$ has conditional expectation 1, so the weighted portfolio has conditional expectation 1.

## AUXILIARY LEMMAS

We collect standard technical bounds used repeatedly in the proofs.

**Lemma 4** (Envelope Bounds). *For every $t \geq 2$ and for each threshold index $j$, the following bounds hold:*

1. $|B_t(u_j)| \leq \frac{1}{\sqrt{t\delta}}$

2. $|A_{t-1}(u_j)| \leq \frac{1}{\sqrt{t\delta}}$

3. $|\Delta Z_t(u_j)| \leq \frac{2}{\sqrt{t\delta}}$

*where $\delta := \min_j u_j(1 - u_j) > 0$.*

*Proof of lemma 4.* We prove each claim in turn. First, we bound the innovation term $B_t(u_j) = \frac{\mathbb{1}\{p_t \leq u_j\} - u_j}{\sqrt{t u_j(1-u_j)}}$. The numerator is bounded by $|\mathbb{1}\{p_t \leq u_j\} - u_j| \leq 1$, and the denominator is bounded below by $\sqrt{t u_j(1 - u_j)} \geq \sqrt{t\delta}$. Combining these gives the first result:

$$|B_t(u_j)| = \frac{|\mathbb{1}\{p_t \leq u_j\} - u_j|}{\sqrt{t u_j(1 - u_j)}} \leq \frac{1}{\sqrt{t u_j(1 - u_j)}} \leq \frac{1}{\sqrt{t\delta}}.$$

Next, we bound the predictable term $|A_{t-1}(u_j)|$ using $A_{t-1}(u_j) = \left(\sqrt{\frac{t-1}{t}} - 1\right) Z_{t-1}(u_j)$. Taking the absolute value gives $|A_{t-1}(u_j)| = \left(1 - \sqrt{\frac{t-1}{t}}\right) |Z_{t-1}(u_j)|$. The pre-factor simplifies to $\frac{1}{\sqrt{t}(\sqrt{t}+\sqrt{t-1})}$, and we can establish a uniform bound $|Z_{t-1}(u_j)| \leq \frac{\sqrt{t-1}}{\sqrt{u_j(1-u_j)}}$. Combining these yields:

$$|A_{t-1}(u_j)| \leq \left(\frac{1}{\sqrt{t}(\sqrt{t} + \sqrt{t-1})}\right) \cdot \left(\frac{\sqrt{t-1}}{\sqrt{u_j(1 - u_j)}}\right)$$

$$= \frac{1}{\sqrt{t}\sqrt{u_j(1 - u_j)}} \cdot \left(\frac{\sqrt{t-1}}{\sqrt{t} + \sqrt{t-1}}\right).$$

Since the final fraction is strictly less than 1, we have $|A_{t-1}(u_j)| < \frac{1}{\sqrt{t}\sqrt{u_j(1-u_j)}} \leq \frac{1}{\sqrt{t\delta}}$, which establishes the second claim.

Finally, the bound on the total increment $|\Delta Z_t(u_j)|$ follows directly from the triangle inequality and the previous two results:

$$
\begin{aligned}
|\Delta Z_t(u_j)| &= |A_{t-1}(u_j) + B_t(u_j)| \\
&\leq |A_{t-1}(u_j)| + |B_t(u_j)| \\
&\leq \frac{1}{\sqrt{t\delta}} + \frac{1}{\sqrt{t\delta}} = \frac{2}{\sqrt{t\delta}}.
\end{aligned}
$$

$\square$

**Lemma 5** (Taylor expansion of $\log \varphi_{t,j}(\lambda)$). *For each fixed $j$ and for all $t \geq 1$,*

$$\log \varphi_{t,j}(\lambda) = \frac{\lambda^2}{2t} + r_{t,j}(\lambda),$$

*with the remainder bounded as*

$$|r_{t,j}(\lambda)| \leq \frac{C|\lambda|^3}{t^{3/2}\,\delta^{3/2}},$$

*for some absolute numerical constant $C > 0$.*

*Proof of lemma 5.* Recall that $\log \varphi_{t,j}(\lambda) = \psi_j(\theta)$ where $\theta = \frac{\lambda}{\sqrt{t u_j(1-u_j)}}$ and

$$\psi_j(\theta) := \log\left(u_j e^{(1-u_j)\theta} + (1 - u_j)e^{-u_j\theta}\right).$$

We compute the first two derivatives of $\psi_j(\theta)$ with respect to $\theta$. Let $f(\theta) = u_j e^{(1-u_j)\theta} + (1 - u_j)e^{-u_j\theta}$. Then $\psi_j(\theta) = \log(f(\theta))$.

**First Derivative:** Using the chain rule, $\psi_j'(\theta) = \frac{f'(\theta)}{f(\theta)}$.

$$f'(\theta) = u_j(1 - u_j)e^{(1-u_j)\theta} - (1 - u_j)u_j e^{-u_j\theta}.$$

At $\theta = 0$, we have $f(0) = u_j + (1 - u_j) = 1$ and $f'(0) = u_j(1 - u_j) - u_j(1 - u_j) = 0$. Therefore,

$$\psi_j'(0) = \frac{0}{1} = 0.$$

**Second Derivative:** We have by simple algebra $\psi_j''(\theta) = \frac{f''(\theta)f(\theta) - (f'(\theta))^2}{(f(\theta))^2}$.

$$f''(\theta) = u_j(1 - u_j)^2 e^{(1-u_j)\theta} + (1 - u_j)(-u_j)^2 e^{-u_j\theta}.$$

At $\theta = 0$, we have $f(0) = 1$, $f'(0) = 0$, and

$$\begin{aligned} f''(0) &= u_j(1 - u_j)^2 + (1 - u_j)u_j^2 \\ &= u_j(1 - u_j)\left[(1 - u_j) + u_j\right] \\ &= u_j(1 - u_j). \end{aligned}$$

Therefore, the second derivative at zero is:

$$\psi_j''(0) = \frac{f''(0) \cdot 1 - 0^2}{1^2} = u_j(1 - u_j).$$

By Taylor's theorem with a remainder, $\psi_j(\theta) = \psi_j(0) + \psi_j'(0)\theta + \frac{\psi_j''(0)}{2}\theta^2 + O(\theta^3)$. Substituting the values we just calculated gives:

$$\begin{aligned} \log\varphi_{t,j}(\lambda) = \psi_j(\theta) &= 0 + 0 \cdot \theta + \frac{u_j(1 - u_j)}{2}\theta^2 + O(\theta^3) \\ &= \frac{u_j(1 - u_j)}{2}\left(\frac{\lambda}{\sqrt{tu_j(1 - u_j)}}\right)^2 + r_{t,j}(\lambda) \\ &= \frac{u_j(1 - u_j)}{2}\frac{\lambda^2}{tu_j(1 - u_j)} + r_{t,j}(\lambda) \\ &= \frac{\lambda^2}{2t} + r_{t,j}(\lambda). \end{aligned}$$

The remainder term $r_{t,j}(\lambda)$ is controlled by the third derivative, which gives the claimed result. $\square$

**Lemma 6** (Range bound for per-step gains). *Fix a bound $|\lambda| \leq \Lambda$. Define the per-threshold gain*

$$g_{t,j} := \lambda(\Delta Z_t(u_j) - A_{t-1}(u_j)) - \log\varphi_{t,j}(\lambda) = \lambda B_t(u_j) - \log\varphi_{t,j}(\lambda).$$

*Then for all $t$ and all $i, k$,*

$$|g_{t,i} - g_{t,k}| \leq \frac{K(\Lambda)}{\sqrt{t}}$$

*for some constant $K(\Lambda)$ depending only on $\Lambda$ and $\delta$.*

*Proof of lemma 6.* By Lemma 4, $|\lambda B_t(u_j)| \leq |\lambda|/\sqrt{t\delta}$. By Lemma 5,

$$\left|\log\varphi_{t,j}(\lambda) - \frac{\lambda^2}{2t}\right| \leq \frac{C|\lambda|^3}{t^{3/2}\delta^{3/2}}.$$

Combine these two uniform bounds to get $|g_{t,j}| \leq C_1(\Lambda)/\sqrt{t}$ and hence the pairwise difference is at most $2C_1(\Lambda)/\sqrt{t}$. Set $K(\Lambda) = 2C_1(\Lambda)$. $\square$

## HEDGE REGRET BOUND AND MAPPING TO OUR GAINS

We use the standard exponential-weights (Hedge) regret bound in the gains formulation. A convenient reference and statement is in [1].

**Proposition 7** (Hedge gains-regret; specialized)**.** *Consider $m$ experts and gains $g_{t,j} \in \mathbb{R}$ for $t = 1, \ldots, T$ and $j = 1, \ldots, m$. Run Hedge with learning rate $\eta > 0$ and initial uniform weights $w_0(j) = 1/m$:*

$$w_t(j) = \frac{w_{t-1}(j) \exp(\eta g_{t,j})}{\sum_{k=1}^{m} w_{t-1}(k) \exp(\eta g_{t,k})}.$$

*Let $\widehat{G}_T := \sum_{t=1}^{T} \sum_{j=1}^{m} w_{t-1}(j) g_{t,j}$ be the forecaster's cumulative gain. Suppose that for each $t$ we have $c_t := \max_j g_{t,j} - \min_j g_{t,j}$. Then for any fixed expert $j$,*

$$\widehat{G}_T \geq \sum_{t=1}^{T} g_{t,j} - \frac{\ln m}{\eta} - \frac{\eta}{8} \sum_{t=1}^{T} c_t^2.$$

*Proof of proposition 7 (Sketch).* This is the standard inequality (see Section 2.2, Theorem 2.2 in [1]). The inequality is derived from convexity of the log-partition function and the elementary inequality $\log(\sum_j w e^{\eta g_{t,j}}) \leq \eta \sum_j w g_{t,j} + \eta^2 c_t^2/8$, which holds when gains are bounded in a range of length $c_t$. The additive $\ln m/\eta$ arises from the initial entropic term. In our application $g_{t,j}$ are the per-step gains defined in previous sections; Lemma 6 gives that $c_t \leq K/\sqrt{t}$, so $\sum_t c_t^2$ behaves like $K^2 \sum_t 1/t = K^2(\log T + O(1))$. $\square$

**Choice of $\eta$.**  To balance the terms $\ln m/\eta$ and $(\eta/8) \sum_t c_t^2$ choose $\eta \asymp \sqrt{\frac{8 \ln m}{K^2 \log T}}$ which yields a regret of order $\lambda \sqrt{\ln m \, \log T}$ after accounting for $K$ depending on $\lambda$.

## PROOF OF THE UNIFIED POWER LOWER BOUND

We restate and prove Theorem 2 in a self-contained way.

**Restated claim.**  Fix a bounded predictable sequence $\{\lambda_t\}$ (or constant $\lambda_t \equiv \lambda$). Let $M_t$ be defined as in (5) with exponential-weights $w_{t-1}$ (any predictable weighting rule). Let $s_t := \mathbb{E}_{H_1}[\Delta Z_t(u^*) \mid \mathcal{F}_{t-1}]$ be the predictable per-step signal for some fixed threshold $u^* = u_{j^*}$. Let $R_T$ denote the deterministic regret bound for the exponential-weights procedure run on gains $g_{t,j}$ (as in Proposition 7). Then for every $T \geq 2$,

$$\mathbb{E}_{H_1}[\log M_T] \geq \sum_{t=1}^{T} \lambda_t s_t - \frac{1}{2} \sum_{t=1}^{T} \frac{\lambda_t^2}{t} - R_T + O(1),$$

where the $O(1)$ term is a convergent series collecting third-order mgf remainders.

*Proof of theorem 2.* Start from the identity

$$\log M_T = \sum_{t=1}^{T} \log G_t = \sum_{t=1}^{T} \log \left( \sum_{j=1}^{m} w_{t-1}(u_j) \exp(g_{t,j}) \right).$$

The logarithm of the aggregated martingale, $\log M_T$, can be bounded from below. We recognize that $M_T$ is a convex combination of exponential processes, with weights $w_{t-1}(u_j)$. Applying Jensen's inequality with the concave logarithm function allows us to move the logarithm inside the weighted sum, yielding:

$$\log M_T \geq \sum_{t=1}^{T} \sum_{j=1}^{m} w_{t-1}(u_j) g_{t,j} =: \widehat{G}_T.$$

Take expectation under $H_1$:

$$\mathbb{E}_{H_1}[\log M_T] \geq \mathbb{E}_{H_1}[\widehat{G}_T].$$

Apply the Hedge regret bound (Proposition 7) to compare $\widehat{G}_T$ to the cumulative gain of the best fixed threshold $j^*$:

$$\widehat{G}_T \geq \sum_{t=1}^{T} g_{t,j^*} - R_T,$$

where $R_T = \dfrac{\ln m}{\eta} + \dfrac{\eta}{8} \sum_t c_t^2$. Taking expectations:

$$\mathbb{E}_{H_1}[\widehat{G}_T] \geq \sum_{t=1}^{T} \mathbb{E}_{H_1}[g_{t,j^*}] - R_T.$$

Now compute $\mathbb{E}_{H_1}[g_{t,j^*}]$. By definition $g_{t,j^*} = \lambda_t B_t(u^*) - \log \varphi_{t,j^*}(\lambda_t)$. Thus

$$\mathbb{E}_{H_1}[g_{t,j^*}] = \lambda_t \, \mathbb{E}_{H_1}[B_t(u^*)] - \log \varphi_{t,j^*}(\lambda_t).$$

By definition $\mathbb{E}_{H_1}[B_t(u^*)] = s_t'$, where $s_t'$ differs from $s_t = \mathbb{E}_{H_1}[\Delta Z_t(u^*) \mid \mathcal{F}_{t-1}]$ by the predictable term $A_{t-1}(u^*)$. Concretely,

$$s_t = \mathbb{E}_{H_1}[\Delta Z_t(u^*) \mid \mathcal{F}_{t-1}] = A_{t-1}(u^*) + \mathbb{E}_{H_1}[B_t(u^*) \mid \mathcal{F}_{t-1}],$$

and after taking unconditional expectation the distinction disappears if we include $A_{t-1}$ inside $s_t$. For simplicity of presentation we use the predictable form $s_t = \mathbb{E}_{H_1}[\Delta Z_t(u^*) \mid \mathcal{F}_{t-1}]$ and note that $\mathbb{E}_{H_1}[B_t(u^*)] = \mathbb{E}_{H_1}[s_t - A_{t-1}(u^*)]$. Rearranging yields

$$\mathbb{E}_{H_1}[g_{t,j^*}] = \lambda_t s_t - \lambda_t A_{t-1}(u^*) - \log \varphi_{t,j^*}(\lambda_t).$$

Gathering terms and summing in $t$,

$$\sum_{t=1}^{T} \mathbb{E}_{H_1}[g_{t,j^*}] = \sum_{t=1}^{T} \lambda_t s_t - \sum_{t=1}^{T} \lambda_t A_{t-1}(u^*) - \sum_{t=1}^{T} \log \varphi_{t,j^*}(\lambda_t).$$

The middle term $\sum_t \lambda_t A_{t-1}$ is small because $A_{t-1} = O(1/\sqrt{t})$ and $\lambda_t$ is bounded — it contributes an $O(1)$ or $O(\lambda\sqrt{\log T})$ term depending on precise assumptions; in our canonical regimes it can be absorbed into lower-order terms. For the last term, apply Lemma 5 to expand:

$$\sum_{t=1}^{T} \log \varphi_{t,j^*}(\lambda_t) = \frac{1}{2} \sum_{t=1}^{T} \frac{\lambda_t^2}{t} + \sum_{t=1}^{T} r_{t,j^*}(\lambda_t),$$

where $\sum_t r_{t,j^*}(\lambda_t)$ converges when $\lambda_t$ is bounded and the cubic remainders are $O(t^{-3/2})$. Combining all parts and reorganizing yields

$$\mathbb{E}_{H_1}[\log M_T] \geq \sum_{t=1}^{T} \lambda_t s_t - \frac{1}{2} \sum_{t=1}^{T} \frac{\lambda_t^2}{t} - R_T + O(1),$$

which is the claimed inequality. $\qquad\square$

## PROOF: ADAPTIVE EXPLOSIVE GROWTH — WEAK REGIME THEOREM 3

**Context.** In the previous section we established the *Unified Power Lower Bound* (theorem 2), which provides a general inequality for the expected log-growth of the e-HC wealth process under the alternative. This result holds without committing to a particular signal regime, and applies broadly to both strong and weak signals.

We now focus on the *weak regime*, corresponding to the rare/weak mixture model where the per-step signal contribution is of order

$$s_t \asymp \frac{\epsilon_t \mu_t \sqrt{n_t}}{\sqrt{t}},$$

with $\epsilon_t \ll 1$ the sparsity parameter and $\mu_t$ the signal strength. In this regime the cumulative signal satisfies

$$S_w(T) = \sum_{t=1}^{T} \frac{\epsilon_t \mu_t \sqrt{n_t}}{\sqrt{t}},$$

which grows only at the $\sqrt{T}$ rate for constant $(\epsilon_t, \mu_t, n_t)$, making detection particularly delicate.

Our objective in what follows is to specialize the unified lower bound to this regime, control all remainder terms explicitly, and strengthen the expectation bound into an *almost sure explosive growth guarantee*. This establishes the weak-regime stopping time result (Corollary 3 in the main text).

ASSUMPTIONS (WEAK REGIME)

Fix a threshold index $j^*$ with $u^* = u_{j^*}$. Let $\delta_0 := u^*(1 - u^*) > 0$ (note $\delta_0 \geq \delta$). Assume that under $H_1$ the one-step increase in the marginal exceedance probability satisfies, for every $t \geq 1$,

$$q_{t,j^*} - u^* \geq c_1 \epsilon_t \mu_t \sqrt{n_t} - c_2 \epsilon_t (\mu_t \sqrt{n_t})^3, \tag{6}$$

with constants $c_1 > 0$, $c_2 \geq 0$. Here $q_{t,j} := \mathbb{P}_{H_1}(p_t \leq u_j)$.

Define the (weak) cumulative signal

$$S_w(T) := \sum_{t=1}^{T} \frac{\epsilon_t \mu_t \sqrt{n_t}}{\sqrt{t}}.$$

We will show a lower bound on $\mathbb{E}_{H_1}[\log M_T]$ of the form

$$\mathbb{E}_{H_1}[\log M_T] \geq \lambda C_0 S_w(T) - \frac{\lambda^2}{2} \log T - C_1 \lambda \sqrt{\ln m \, \log T} - C_2,$$

for explicit constants $C_0, C_1, C_2$ (depending on $u^*$, $\delta$, and algorithmic parameters like the Hedge constant $K$). The derivation optimizes the Hedge learning rate and controls all remainder terms.

STEP 1: START FROM UNIFIED BOUND

By the result of Section 11 (unified power theorem) we have, for a fixed bounded $\lambda$ (we take $\lambda_t \equiv \lambda$),

$$\mathbb{E}_{H_1}[\log M_T] \geq \lambda \sum_{t=1}^{T} s_t - \frac{\lambda^2}{2} \sum_{t=1}^{T} \frac{1}{t} - R_T + O(1),$$

where $s_t = \mathbb{E}_{H_1}[\Delta Z_t(u^*) \mid \mathcal{F}_{t-1}]$ is the predictable per-step signal and $R_T$ is the Hedge regret.

STEP 2: RELATE $s_t$ TO THE MIXTURE PARAMETERS (WEAK REGIME)

By definition,
$$\Delta Z_t(u^*) = A_{t-1}(u^*) + B_t(u^*),$$
hence
$$s_t = \mathbb{E}_{H_1}[\Delta Z_t(u^*) \mid \mathcal{F}_{t-1}] = A_{t-1}(u^*) + \mathbb{E}_{H_1}[B_t(u^*) \mid \mathcal{F}_{t-1}].$$
Taking unconditional expectation and using $\mathbb{E}[A_{t-1}] = O(1/\sqrt{t})$ gives
$$\mathbb{E}_{H_1}[s_t] = \mathbb{E}_{H_1}[B_t(u^*)] + O(1/\sqrt{t}).$$
Now compute $\mathbb{E}_{H_1}[B_t(u^*)] = \dfrac{q_{t,j^*} - u^*}{\sqrt{t \, u^*(1 - u^*)}}$. Using assumption (6),

$$\mathbb{E}_{H_1}[B_t(u^*)] \geq \frac{c_1 \epsilon_t \mu_t \sqrt{n_t}}{\sqrt{t \, u^*(1 - u^*)}} - \frac{c_2 \epsilon_t (\mu_t \sqrt{n_t})^3}{\sqrt{t \, u^*(1 - u^*)}}.$$

Summing this expression from $t = 1$ to $T$ gives:

$$\sum_{t=1}^{T} \mathbb{E}_{H_1}[B_t(u^*)] \geq \sum_{t=1}^{T} \frac{c_1 \epsilon_t \mu_t \sqrt{n_t}}{\sqrt{t \, u^*(1 - u^*)}} - \sum_{t=1}^{T} \frac{c_2 \epsilon_t (\mu_t \sqrt{n_t})^3}{\sqrt{t \, u^*(1 - u^*)}}$$

$$= \frac{c_1}{\sqrt{u^*(1 - u^*)}} \sum_{t=1}^{T} \frac{\epsilon_t \mu_t \sqrt{n_t}}{\sqrt{t}} - \text{higher order terms}$$

$$= C_0 S_w(T) - \text{h.o.t.}$$

where we define $C_0 := c_1/\sqrt{u^*(1-u^*)}$ and $S_w(T) := \sum_{t=1}^{T} \frac{\epsilon_t \mu_t \sqrt{n_t}}{\sqrt{t}}$. The full sum of the expected signal is then:

$$\sum_{t=1}^{T} \mathbb{E}_{H_1}[s_t] = \sum_{t=1}^{T} \mathbb{E}_{H_1}[A_{t-1}(u^*)] + \sum_{t=1}^{T} \mathbb{E}_{H_1}[B_t(u^*)]$$

$$\geq \left( \sum_{t=1}^{T} \mathbb{E}_{H_1}[A_{t-1}(u^*)] \right) + C_0 S_w(T) - \text{h.o.t.}$$

Since $|A_{t-1}|$ is $O(1/\sqrt{t})$, its sum is absorbed into the lower-order terms, leading to the final bound $\sum \mathbb{E}_{H_1}[s_t] \geq C_0 S_w(T) - \widetilde{C}_2$.

STEP 3: BOUND THE REGRET $R_T$ AND CHOOSE $\eta$

From Proposition 7 the regret satisfies

$$R_T \leq \frac{\ln m}{\eta} + \frac{\eta}{8} \sum_{t=1}^{T} c_t^2,$$

with $c_t \leq K(\Lambda)/\sqrt{t}$ by Lemma 6. Thus

$$\sum_{t=1}^{T} c_t^2 \leq K(\Lambda)^2 \sum_{t=1}^{T} \frac{1}{t} = K(\Lambda)^2 (\log T + O(1)).$$

Choosing

$$\eta = \sqrt{\frac{8 \ln m}{K(\Lambda)^2 \log T}}$$

balances the two terms and gives

$$R_T \leq C_1(\Lambda) \, \lambda \sqrt{\ln m \, \log T} + O(1),$$

for some constant $C_1(\Lambda)$ (note $K(\Lambda)$ depends on $\lambda$ via Lemma 6, hence we write dependence on $\Lambda$).

STEP 4: ASSEMBLE THE PIECES

Plugging the lower bound for $\sum_t \mathbb{E}_{H_1}[s_t]$ and the regret bound into the unified-power inequality in Section 11 yields

$$\mathbb{E}_{H_1}[\log M_T] \geq \lambda C_0 S_w(T) - \frac{\lambda^2}{2} \log T - C_1(\Lambda) \, \lambda \sqrt{\ln m \, \log T} - C_2,$$

with $C_2$ absorbing $\widetilde{C}_2$ and other $O(1)$ terms (including the convergent cubic mgf remainders from Lemma 5). This is exactly the claimed bound.

STEP 5: ALMOST-SURE EXPLOSIVE GROWTH

To upgrade the growth in expectation to an almost-sure result, we must show that the stochastic fluctuations of $\log M_T$ around its mean are well-controlled. We write $\log M_T$ as a sum of one-step gains:

$$\log M_T \geq \sum_{t=1}^{T} \widehat{g}_t \quad \text{where} \quad \widehat{g}_t := \sum_{j=1}^{m} w_{t-1}(u_j) g_{t,j}.$$

Let $\zeta_t = \widehat{g}_t - \mathbb{E}_{H_1}[\widehat{g}_t \mid \mathcal{F}_{t-1}]$. The sequence $\{\zeta_t\}_{t \geq 1}$ is a martingale difference sequence adapted to the filtration $\{\mathcal{F}_t\}$. The total fluctuation of our gain process around its conditional expectation is the martingale $\mathcal{Z}_T = \sum_{t=1}^{T} \zeta_t$.

We apply a standard concentration inequality for martingales, such as Freedman's inequality.

**Freedman's Inequality.** Let $\{X_t\}_{t=1}^T$ be a real-valued martingale difference sequence with respect to a filtration $\{\mathcal{F}_t\}$. Assume there is a uniform bound $|X_t| \leq K$ for all $t$. Let $V_T = \sum_{t=1}^T \mathbb{E}[X_t^2 \mid \mathcal{F}_{t-1}]$ be the predictable quadratic variation. Then for any $x > 0$:

$$\mathbb{P}\left(\sum_{t=1}^T X_t \geq x\right) \leq \exp\left(-\frac{x^2}{2(V_T + Kx)}\right).$$

**Mapping to Our Problem.** In our case, the martingale difference is $X_t = \zeta_t$.

1. **Uniform Bound (K):** By Lemma 3 (Envelope Bounds), we know that for a fixed $\lambda$, $|g_{t,j}| \leq C(\lambda)/\sqrt{t}$ for some constant $C(\lambda)$. Since $\widehat{g}_t$ is a convex combination of the $g_{t,j}$, it is also bounded by $|\widehat{g}_t| \leq C(\lambda)/\sqrt{t}$. The same applies to its conditional expectation. Therefore, $|\zeta_t| \leq 2C(\lambda)/\sqrt{t}$. The largest bound occurs at $t = 1$, so we can set the uniform bound $K = 2C(\lambda)$.

2. **Predictable Variation ($V_T$):** We can bound the conditional variance:

$$\mathbb{E}[\zeta_t^2 \mid \mathcal{F}_{t-1}] = \mathrm{Var}(\widehat{g}_t \mid \mathcal{F}_{t-1}) \leq \mathbb{E}[\widehat{g}_t^2 \mid \mathcal{F}_{t-1}] \leq \mathbb{E}[(C(\lambda)/\sqrt{t})^2 \mid \mathcal{F}_{t-1}] = \frac{C(\lambda)^2}{t}.$$

Summing this gives an upper bound on the predictable variation:

$$V_T = \sum_{t=1}^T \mathbb{E}[\zeta_t^2 \mid \mathcal{F}_{t-1}] \leq C(\lambda)^2 \sum_{t=1}^T \frac{1}{t} = O(\log T).$$

Now, let's analyze the probability of the fluctuations being large and negative. Applying Freedman's inequality to the sequence $\{-\zeta_t\}$ (which is also a martingale difference sequence), we get:

$$\mathbb{P}\left(\sum_{t=1}^T \zeta_t \leq -x\right) = \mathbb{P}\left(\sum_{t=1}^T (-\zeta_t) \geq x\right) \leq \exp\left(-\frac{x^2}{2(V_T + Kx)}\right).$$

Let's choose $x = \epsilon S_w(T)$ for some small $\epsilon > 0$. Under the weak-regime assumption that $S_w(T) \gg \log T$, the denominator is dominated by the $Kx$ term for large $T$, and the probability of a large negative deviation decays exponentially.

More formally, the condition $S_w(T) \to \infty$ with $\log T = o(S_w(T))$ ensures that the expected growth $\mathbb{E}[\log M_T] \approx \lambda C_0 S_w(T)$ grows much faster than the scale of the random fluctuations, which is of order $\sqrt{V_T} \approx \sqrt{\log T}$. By the Borel-Cantelli lemma, this ensures that $\log M_T \to \infty$ almost surely. Therefore, $M_T \to \infty$ almost surely, and the test, which rejects when $M_t \geq 1/\alpha$, will detect the signal with probability one. □

EXPECTED STOPPING-TIME: HIGH-PROBABILITY BOUND AND EXPECTATION

Let $\tau := \inf\{t \geq 1 : M_t \geq 1/\alpha\}$ be the stopping time (the test stops when wealth crosses threshold $1/\alpha$). We give a high-probability finite-time guarantee and then convert it to an expectation bound via a tail-sum.

**Theorem 8** (High-probability finite-time detection and expectation bound). *Under the assumptions of the weak-regime theorem and the same notation, fix any $T \geq 2$. There exist constants $C_0, C_1, C_2$ such that with probability at least $1 - \delta$ under $H_1$,*

$$\tau \leq T \quad \text{whenever} \quad \lambda C_0 S_w(T) - \frac{\lambda^2}{2}\log T - C_1\lambda\sqrt{\ln m \ \log T} - C_2 \geq \log(1/\alpha) + \sqrt{2V_T \log(1/\delta)} + b\log(1/\delta),$$

*where $V_T$ is an upper bound on the predictable variance of the martingale fluctuations of $\log M_T$ and $b$ is a uniform bound on single-step increments of the martingale difference sequence. Consequently, choosing a sequence $T(\delta)$ that satisfies the displayed inequality yields $\mathbb{P}_{H_1}(\tau \leq T(\delta)) \geq 1 - \delta$, and the expected stopping time can be bounded by*

$$\mathbb{E}_{H_1}[\tau] \leq \sum_{k \geq 0} T(2^{-k}).$$

*Sketch and explanation of the inequality.* We start from the deterministic lower bound on the expectation:

$$\mathbb{E}_{H_1}[\log M_T] \geq \Lambda_T := \lambda C_0 S_w(T) - \frac{\lambda^2}{2} \log T - C_1 \lambda \sqrt{\ln m \ \log T} - C_2.$$

We wish to control the lower tail of $\log M_T$. Write $\log M_T = \mathbb{E}_{H_1}[\log M_T] + \Delta_T$ where $\Delta_T$ is the mean-zero fluctuation (a sum of martingale differences). Apply Freedman's inequality (one-sided form) to $\Delta_T$: for any $x > 0$,

$$\mathbb{P}\left(\Delta_T \leq -x\right) \leq \exp\left(-\frac{x^2}{2(V_T + bx)}\right),$$

where $V_T$ is an upper bound on the predictable variance of the increments and $b$ bounds their absolute size. Set the right-hand side equal to $\delta$ and solve for $x$ to get a tail bound of the desired form $x \lesssim \sqrt{2V_T \log(1/\delta)} + b \log(1/\delta)$. Therefore with probability at least $1 - \delta$,

$$\log M_T \geq \Lambda_T - \left(\sqrt{2V_T \log(1/\delta)} + b \log(1/\delta)\right).$$

Hence if $\Lambda_T - (\cdots) \geq \log(1/\alpha)$ then $M_T \geq 1/\alpha$ with probability at least $1 - \delta$, which implies $\tau \leq T$ with probability at least $1 - \delta$.

To bound the expectation $\mathbb{E}[\tau]$, use the tail-sum formula for nonnegative integer-valued random variables:

$$\mathbb{E}[\tau] = \sum_{t \geq 0} \mathbb{P}(\tau > t).$$

Choose dyadic levels $T(2^{-k})$ that achieve high probability $1 - 2^{-k}$ detection (via the displayed inequality with $\delta = 2^{-k}$). Then $\sum_k T(2^{-k})$ upper bounds $\mathbb{E}[\tau]$. In many regimes (e.g., weak constant-parameter with $S_w(T) \asymp c\sqrt{T}$) this yields the heuristic scaling $\mathbb{E}[\tau] \asymp (\log(1/\alpha)/(\lambda c))^2$. Making this precise requires computing $V_T$ (which is of order $\lambda^2 \log T$) and $b$ (order $\lambda/\sqrt{1}$), giving the form appearing in the theorem statement. $\qquad\square$

**Remarks.** This argument is standard in sequential detection: combine an expectation lower bound for a growable statistic with concentration of martingale fluctuations to convert the expectation bound into a high-probability stopping-time guarantee, then convert the high-probability guarantee into an expectation bound by summation of tails. The arguments above are non-asymptotic and explicit once the constants $V_T$ and $b$ are computed (they depend on $\lambda$ and $\delta$ from Lemma 6).

## TECHNICAL REMARKS AND CONSTANTS SUMMARY

- The constants $C_0, C_1, C_2$ appearing in the weak-regime theorem are explicit in the proof: $C_0 = c_1/\sqrt{u^*(1 - u^*)}$, $C_1$ depends on the range constant $K(\Lambda)$ from Lemma 6 and thus on $\lambda$ and $\delta$, and $C_2$ collects cubic mgf remainders and the deterministic $A_{t-1}$-contributions.

- The Hedge regret mapping used a learning rate $\eta$ tuned to balance $\ln m/\eta$ and $\eta \sum_t c_t^2$; any data-dependent or second-order Hedge variant that exploits observed $c_t$ can reduce the $\sqrt{\ln m \ \log T}$ term and improve practical performance.

- Our results hold for predictable $\lambda_t$ (possibly data-dependent), provided $|\lambda_t|$ is uniformly bounded so that the Taylor remainder series and range bounds remain controlled.

- For stronger asymptotic clarity, in the weak regime with fixed $\epsilon, \mu, n$ we have $S_w(T) \asymp \epsilon \mu \sqrt{T}$, so the leading term in the expected log-wealth is $\lambda C_0 \epsilon \mu \sqrt{T}$, while the negative term is $(\lambda^2/2) \log T$. Thus explosive growth occurs as soon as $\epsilon \mu \sqrt{T} \gg \log T$.

