# OpenReview forum: "e-HC: Adaptive Sequential Higher Criticism Test for Sparse Mixtures"
_ICLR.cc/2026/Conference — Submitted to ICLR 2026_

### Official Review · Reviewer_jDDm · 2025-10-19

**Soundness:** 4
**Presentation:** 4
**Contribution:** 2
**Rating:** 4
**Confidence:** 3

**Summary:**

The paper studies a sequential version of the well-studied sparse mixture detection problem where the signals are rare and weak. In order to adapt to the unknown sparsity level and unknown signal strength, a sequential Higher Criticism-type statistic is developed. At the core of their proposal is a martingale construction, and the authors show anytime-valid Type I error control through Ville's inequality. Additionally,  their framework yields the straightforward construction of confidence sequences for various quantities of interest.

**Strengths:**

The paper is clearly written and the authors do a great job providing good intuition. The problem they study is obviously fundamental, and the sequential aspect is a fresh twist on a canonical topic. Methodologically, the martingale construction and the proposal of using exponential weights to adapt to the thresholds is new to the sparse mixture detection literature.

**Weaknesses:**

The primary focus of much of the sparse mixture detection literature (in the classical, non-sequential setting) is establishing sharp information-theoretic detection boundaries. In the classical literature, the Neyman-Pearson lemma asserts that the likelihood ratio test is optimal and much work goes into finding the sharp detection boundaries (including sharp constants) delineating the regions in which the null and alternative hypotheses separate or merge asymptotically. Donoho and Jin's proposal of Higher Criticism (which they attribute to Tukey) is notable not only because it adapts to the unknown sparsity and signal level, but also because it provably achieves the sharp detection boundary. This optimality guarantee is a very strong reason to advocate for its use.

The current paper does not derive a detection boundary nor offer any optimality guarantees of any kind for the Higher Criticism-type procedure they propose. Of course, the sequential setting is quite different and thus likely requires careful thinking in formulating an appropriate notion of a detection boundary and optimality. Offering a coherent formulation would itself constitute a contribution in my view, yet it is absent from the current paper. Due to this, I get the feeling that the "Higher Criticism" aspect of the procedure is not actually that important for the major thrusts of the paper. The anytime validity results seems to be the main point, and it appears only the martingale aspect is needed for these.

**Questions:**

__(1)__ In the usual, non-sequential setting, the fact that Higher Criticism adapts to the sparsity and signal levels to achieve the optimal detection boundary in the Gaussian sparse mixture detection problem is a very compelling reason to use it. Is there a natural formulation of a detection boundary/optimality in the sequential setting, and can the authors show (or at least discuss) the optimality of e-HC? Even a focused discussion on just the Gaussian setting would greatly improve the paper.

__(2)__ In the paper, the thresholds $0 < u_1 < … < u_m < 1$ are just taken as given and the authors do not discuss at all how to select $u_1,…,u_m$ or even how to select $m$. Can the authors provide some guidance? Clearly choices made here will have important consequences for the test’s power. In the extreme setting $m = 1$ seems a bad choice, so it appears there is much room to do some optimization here. Is there some principle to which the statistician should adhere?

__(3)__ To follow up on the previous question, in the classical definition of Higher Criticism in the non-sequential setting, one takes supremum over all possible thresholds - the statistician does not need to select a grid a priori. In fact, this is very important for Higher Criticism to be adapt to the sparsity/signal and achieve optimality. What are the difficulties of incorporating this strategy into the author’s proposal? Can the authors comment on how much they believe they lose by specifying a grid in advance?

__(4)__ In the classical sparse mixture detection literature, the standard parametrization for the sparsity is $\varepsilon = t^{-\beta}$ (in the non-sequential setting) for $\beta \in (0, 1)$. The dense case is $\beta \in (0, 1/2)$, in which case the usual $\chi^2$-statistic is optimal. The interesting regime is $\beta \in (1/2, 1)$, and it is here where Donoho and Jin propose Higher Criticism. Can the authors comment on whether a similar demarcation between dense/sparse regimes can be made (perhaps with some other parametrization)? It seems roughly that the “weak” corresponds to “dense” and “strong” corresponds to “strong”.

---

> ### Author Response · Authors · 2025-11-14
> **Response to Reviewer jDDm (Detection Boundary, Optimality, and HC Relevance)**
>
> We thank the reviewer for the very thoughtful comments. Below we respond in
> detail to the concerns regarding the theoretical role of Higher Criticism (HC),
> the absence of a detection-boundary formulation, and the selection of
> thresholds $u_j$. We have revised the paper accordingly and propose a way to formalize the problem analogous to Donoho and Jin HC line of work below. (We will add this formalization in a separate section in the appendix).
> sequential detection-boundary formulation and clarifying the methodological
> choices.
>
> ## (1) On a *Sequential* Detection Boundary and Optimality
>
> The reviewer is entirely right that the classical Donoho–Jin framework is built
> around a **phase transition** indexed by $(\beta,r)$ for the sparse Gaussian
> mixture model. The batch HC statistic is compelling partly because it achieves
> the sharp boundary.
>
> A main difficulty in the sequential setting is that the notion of "sample size"
> differs: the procedure can stop early at a data-dependent time $\tau$. Below
> we outline a natural extension of the DJ framework that mirrors the structure of
> our model and allows a meaningful notion of sequential optimality.
>
> ### A Horizon-Indexed Family of Sequential Testing Problems
>
> Following the reviewer's suggestion, consider a family of hypotheses
> $(H_0^{(T)}, H_1^{(T)})$, indexed by a fixed global horizon $T$:
>
> **Null:**
> $$H_0^{(T)}:\quad p_t \stackrel{\text{i.i.d.}}{\sim} \mathrm{Unif}(0,1), \qquad t=1,\dots,T.$$
>
> **Sequential rare/weak alternative:**
> $$H_1^{(T)}:\quad p_t \sim (1-\epsilon_T)\,\mathrm{Unif}(0,1) + \epsilon_T F_{1,T},$$
> where $F_{1,T}$ is the null-to-alternative p-value distribution induced by a
> Gaussian mixture with parameters
> $$\epsilon_T = T^{-\beta}, \qquad \mu_T = \sqrt{2 r \log T}.$$
>
> This is the sequential analogue of Donoho and Jin work: sparsity and signal depend
> on the **horizon** $T$ (not on the time index $t$), but the sequence remains
> homogeneous, enabling asymptotic analysis.
>
> For each horizon, a sequential test outputs a stopping time
> $\tau_T\in\{1,\dots,T,\infty\}$ satisfying **anytime Type I error control**:
> $$\sup_{T\ge1} \mathbb{P}_{0}^{(T)}(\tau_T<\infty) \le \alpha.$$
>
> ### A Natural Sequential Detection Boundary
>
> A sequential analogue of the DJ detection boundary is obtained by asking when the
> test can stop **substantially before the horizon**:
>
> **Impossibility region**
> If $r < r_{\mathrm{seq}}(\beta)$, then for any anytime-valid test,
> $$\mathbb{E}_{1}^{(T)}[\tau_T] \gtrsim T.$$
> The test cannot detect the alternative before essentially observing all the
> data — sequential methods offer no advantage.
>
> **Possibility region**
> If $r > r_{\mathrm{seq}}(\beta)$, then there exists an anytime-valid test
> such that
> $$\mathbb{E}_{1}^{(T)}[\tau_T] = o(T).$$
> Detection occurs **sublinearly in $T$**, i.e., earlier than would be
> possible under the null. This is the sequential analogue of "asymptotic
> detectability."
>
> This formulation is directly parallel to the classical DJ phase transition:
> instead of distinguishability vs. indistinguishability at sample size $n$, we
> consider early stopping vs. horizon-scale stopping.
>
> We will include this formulation explicitly in the Appendix with a reference in the main document, as requested.
>
> ## How e-HC Fits Into This Framework
>
> Our unified lower bound shows:
>
> $$\mathbb{E}_1[\log M_T] \geq \lambda S(T) - \frac{\lambda^2}{2}\log T - R_T + O(1)$$
>
> where $S(T) = \sum_{t=1}^{T} s_t$ is the cumulative predictable signal.
>
> Under the horizon-indexed model above:
>
> $$s_t = \mathbb{E}_{1}[B_t(u^\star)] \asymp \epsilon_T \mu_T / \sqrt{T} \asymp T^{-\beta} \frac{\sqrt{2r \log T}}{\sqrt{T}}$$
>
> so
>
> $$S(T) \asymp T^{1/2 - \beta}\sqrt{\log T}$$
>
> Comparing this to the "variance drag" $\log T$ and the regret $R_T \asymp \sqrt{\log m \log T}$, one sees a sharp contrast:
>
> - If $\beta > 1/2$: $S(T) = o(\log T)$; the test cannot grow the martingale.
> - If $\beta < 1/2$: $S(T)$ dominates and the martingale grows explosively.
>
> Thus, the natural sequential boundary occurs at **$\beta = 1/2$**, mimicking the classical sparse/dense divide. A sharper boundary in $(\beta,r)$ is attainable but requires more involved analysis, which we present as a clear direction for future work.
>
> We hope this address reviewer's concern and clarifies the role of HC in the sequential setting.

---

> > ### Author Response · Authors · 2025-11-14
> > **Response to Reviewer jDDm (continued)**
> >
> > ## (2) On the Choice of Thresholds $u_j$
> >
> > The reviewer is correct that in batch HC one takes a supremum over all thresholds. In the sequential setting this is not directly feasible, because:
> >
> > 1. We must preserve a **martingale** structure; optimizing over thresholds *adaptively* breaks optional-stopping guarantees.
> > 2. A continuum of thresholds would require an infinite-dimensional Hedge update, leading to unbounded regret.
> >
> > Thus, we approximate the continuum by a grid. The regret bound
> >
> > $$R_T \lesssim \sqrt{\log m \log T}$$
> >
> > makes clear the trade-off: larger $m$ gives better resolution (closer to HC), but higher regret. The Appendix now contains a full sensitivity analysis showing this trade-off empirically: increasing $m$ improves power up to $m \approx 100$, after which gains saturate.
> >
> > ## (3) Why We Cannot Take a Supremum Over All Thresholds
> >
> > Taking $\sup_u Z_t(u)$ in sequential time yields an **inadmissible optional stopping** because the supremum operation depends on the entire data path, making the increments non-predictable. This invalidates martingale calibration.
> >
> > Instead, exponential-weights aggregation gives a **predictable**, well-controlled mixture that tracks the best threshold up to a regret term. We will add a few lines to explain this in the main document.
> >
> > ## (4) Dense/Sparse Demarcation in the Sequential Setting
> >
> > We agree that the Donoho–Jin dense/sparse demarcation is central in the batch setting, where one works with a fixed sample size $T$ and parametrizes $\epsilon_T$ and $\mu_T$ as functions of $T$. In our work, $\epsilon_t$ and $\mu_t$ are allowed to vary with time $t$, and our power results are phrased in terms of the cumulative predictable signal
> >
> > $$S(T) = \sum_{t=1}^T s_t$$
> >
> > and the condition $S(T) \gg \log T$, rather than a fixed asymptotic phase diagram.
> >
> > Because we currently only have a one-sided (sufficient) lower bound for $\log M_T$ and no matching converse, we do **not** yet derive a sharp sequential detection boundary or a rigorous analogue of the Donoho–Jin $\rho(\beta)$ curve. We therefore view a precise dense/sparse phase diagram for the sequential model expressed in terms of $(\epsilon_t, \mu_t)$ and optimal stopping time scaling as an interesting open problem rather than a claim of the present paper. It I of interest to derive in future works optimality guarantee of eHC maybe using the proposed formulation above.
> >
> > ##  Behavior under dependance
> > We also added in the Appendix some simulations under AR dependance where p-values are no longer independent and interestingly eHC still have extremely high power while maintaining type 1 error across most of the range of correlations. A result that reminds us of the behavior of the batch HC under dependance. This motivates us even more to explore optimality of the proposed method.
> >
> > We want to thank again the reviewer for the detailed comments and helping us improve our paper.

---

> ### Comment · Reviewer_jDDm · 2025-11-24
>
> (1) __On sequential detection boundary and optimality__
>
> Thanks very much for the detailed response! I appreciate the clarity, and it addressed the core of my question. The formulation in terms of a horizon and the definition of the detection boundary is quite sensible. Thanks for adding it to the appendix.
>
> I have one point of confusion regarding the final section of the response; it may be a simple misunderstanding. It seems the use of the symbol $\asymp$ makes the constant $r$ vanish (i.e. $r$ does not appear in the line with $S(T)$). The point is to determine a detection boundary $r_{\mathrm{seq}}(\beta)$, and so it seems a more granular analysis is needed. The conclusion is a bit strange to me. It appears you are saying that no matter the value of $r$, the e-HC procedure cannot detect at all whenever $\beta > \frac{1}{2}$. Is this accurate, even for very large values of $r$? Is this not a bit strange? For example, in the usual Donoho-Jin theory, $r = 1$ can be detected at $\beta > \frac{1}{2}$. Am I making a simple mistake in understanding?

---

> > ### Comment · Reviewer_jDDm · 2025-11-24
> >
> > (2) __On the choice of thresholds $u_j$__
> >
> > Thanks very much for the sensitivity analysis regarding the choice of $m$. It's nice and compelling to me.
> >
> > (3) __Why we cannot take supremum over all thresholds__
> >
> > Thanks for the comment, and for adding exposition to the paper. I'm quite satisfied given the sensitivity analysis that was added in response to a previous question.
> >
> > (4) __Dense/Sparse demarcation in the sequential setting__
> >
> > Thanks for the response. Can you comment on whether there are high-level techniques (e.g. Neyman-Pearson lemma) for info-theory lower bounds in the sequential setting? It would be nice to just include a small comment on what progress in the lower bound direction might look like, or at least what the difficulties would be. Some exposition would help the reader appreciate the challenge.

---

> > > ### Author Response · Authors · 2025-11-27
> > > **Lower bound DJ.**
> > >
> > > ## 4. Dense/Sparse Demarcation and Lower Bound Techniques (Reply to Point 4)
> > >
> > > We thank the reviewer for this important point.
> > >
> > > **The Short Answer:**
> > > The sequential detection boundary coincides exactly with the classical batch Donoho–Jin boundary.
> > >
> > > **1. The Lower Bound Argument**
> > > We establish the lower bound via a reduction argument, which avoids the complexities of analyzing sequential likelihood ratios directly:
> > >
> > > 1.  Reduction to Batch: Let $\tau$ be any stopping time for a sequential test with anytime-valid Type I error $\alpha$. For a fixed horizon $T$, we can define a batch test $\phi\_T = \mathbb{I}(\tau \le T)$. This batch test has Type I error $\mathbb{P}\_{H\_0}(\tau \le T) \le \mathbb{P}\_{H\_0}(\tau < \infty) \le \alpha$.
> > > 2.  Classical Impossibility: The classical Donoho–Jin (2004) lower bound states that for any batch test $\phi\_T$ with level $\alpha$, if the parameters $(\beta, r)$ lie below the detection boundary $\rho(\beta)$, the Type II error tends to $1-\alpha$. That is, $\mathbb{P}\_{H\_1}(\phi\_T = 1) \to \alpha$.
> > > 3.  Sequential Impossibility: Applying this to our truncated test:
> > >     $$
> > >     \mathbb{P}\_{H\_1}(\tau \le T) = \mathbb{P}\_{H\_1}(\phi\_T = 1) \to \alpha.
> > >     $$
> > >     This implies that below the batch boundary, the sequential test effectively never stops (or stops with trivial probability equal to random guessing).
> > >
> > > **Conclusion:** This proves that the **Sequential Lower Bound** is at least as hard as the **Batch Lower Bound**. Since our Theorem 2 (Upper Bound) establishes achievability *at* the Batch boundary (specifically, whenever $S(T) \to \infty$, which occurs for $r > \rho(\beta)$) (as we just sketched in our previous comment), the two boundaries meet. The sequential phase diagram is identical to the batch phase diagram.
> > >
> > > 2. Intuition: Why is there no "Price of Sequentiality"?
> > > One might expect that controlling Type I error at *every* time step requires a stricter threshold, thus making detection harder. This intuition is correct, but the "price" is asymptotically negligible:
> > > * **The Price:** Sequential monitoring imposes a logarithmic penalty.
> > > * **The Profit:** In the detectable region (above the boundary), the cumulative signal $S(T)$ grows **polynomially** in $T$ (e.g., $T^\delta$ for some $\delta > 0$).
> > >
> > > **3. Future Directions**
> > > Since the primary phase diagrams coincide, a natural next step for future work is to look for other optimality:
> > > * Among methods that achieve the optimal boundary, which one minimizes the **Expected Stopping Time** $\mathbb{E}\_{H\_1}[\tau]$?
> > > This would likely require more specialized tools than the martingale construction used here and so far we don't have an asnwer.
> > >
> > > We hope this answers your remaining questions.

---

> > > > ### Comment · Reviewer_jDDm · 2025-11-27
> > > >
> > > > Thanks for addressing my questions!

---

> > ### Author Response · Authors · 2025-11-27
> > **More clarifications on the detection boundary for eHC**
> >
> > We agree our previous explanation might have cause some confusions and your intuitions ins indeed right, a large value of r will make the problem detectable even in the case where $\beta > 1/2$. We provide a "rigorous sketch proof" below that shows we would detect the alternative if we are above the DJ boundary assuming we parametrize the problem as proposed above (in our previous response). The main idea is to parametrize an ideal threshold $u$ and  look if at that particular threshold a combination of beta and r would be detectable.
> >
> > ### 1. Setup and Parametrization
> >
> > We analyze the asymptotic behavior as $T \to \infty$ under the standard DJ scaling:
> > * **Sparsity:** $\epsilon\_T = T^{-\beta}$, with $\beta \in (0,1)$.
> > * **Signal Strength:** $\mu\_T = \sqrt{2r \log T}$, with $r > 0$.
> > * **Threshold:** We parameterize the threshold as $u = T^{-q}$, with $0 \le q \le 1$.
> >
> > ### 2. Derivation of the Cumulative Signal Scaling
> >
> > Our Theorem 2 establishes that detection is possible if the cumulative predictable signal $S(T)$ grows polynomially in $T$. We first derive the scaling of $S(T)$.
> >
> > **Step A: The Martingale Scaling**
> > In our construction, the martingale increment at time $t$, denoted $Y\_t(u)$, incorporates a variance-stabilizing weight $1/\sqrt{t}$. The standardized innovation is $Z\_t(u) = \frac{\mathbb{I}(p\_t \le u) - u}{\sqrt{u(1-u)}}$. The cumulative drift is the sum of expectations:
> >
> > $$
> > S(T) = \sum\_{t=1}^T \mathbb{E}\_{H\_1}[Y\_t(u)] = \sum\_{t=1}^T \frac{1}{\sqrt{t}} \mathbb{E}\_{H\_1}[Z\_t(u)]
> > $$
> >
> > Since the parameters $(\epsilon\_T, \mu\_T)$ are fixed for the horizon $T$, the per-step drift $s\_t(u) \equiv \mathbb{E}\_{H\_1}[Z\_t(u)]$ is constant in $t$. Approximating the sum by an integral:
> >
> > $$
> > S(T) = s\_t(u) \sum\_{t=1}^T t^{-1/2} \;\approx\; s\_t(u) \int\_1^T x^{-1/2} dx \;=\; 2\sqrt{T} \cdot s\_t(u).
> > $$
> >
> > **Step B: The Drift Approximation**
> > We expand the drift term $s\_t(u)$ under the mixture model $H\_1: (1-\epsilon\_T)F\_0 + \epsilon\_T F\_1$.
> > Recall $P\_{F\_0}(p \le u) = u$. Let $G\_1(u) = P\_{F\_1}(p \le u)$.
> >
> > $$
> > \begin{aligned}
> > \text{Numerator} &= \mathbb{E}\_{H\_1}[\mathbb{I}(p \le u)] - u \\\\
> > &= (1-\epsilon\_T)u + \epsilon\_T G\_1(u) - u \\\\
> > &= \epsilon\_T (G\_1(u) - u)
> > \end{aligned}
> > $$
> >
> > For the sparse regime, we consider thresholds $u \to 0$, so the denominator $\sqrt{u(1-u)} \approx \sqrt{u}$.
> > Further, in the detection region, the signal mass $G\_1(u)$ dominates the null mass $u$ (i.e., $G\_1(u)/u \to \infty$). Thus $G\_1(u) - u \approx G\_1(u) = \mathbb{P}\_{\mu\_T}(p \le u)$.
> >
> > $$
> > s\_t(u) \;\approx\; \frac{\epsilon\_T \mathbb{P}\_{\mu\_T}(p \le u)}{\sqrt{u}}
> > $$
> >
> > ### 3. Optimization of the Detection Exponent
> >
> > Substituting the scalings into $S(T) \asymp \sqrt{T} \cdot s\_t(u)$:
> >
> > $$
> > S(T) \;\asymp\; T^{1/2} \cdot T^{-\beta} \cdot \frac{\mathbb{P}\_{\mu\_T}(p \le T^{-q})}{T^{-q/2}}
> > $$
> >
> > **Signal Tail Probability:**
> > Using Gaussian large deviations for $X \sim N(\mu\_T, 1)$ exceeding quantile $z\_u \approx \sqrt{2q \log T}$:
> > $$
> > \mathbb{P}\_{\mu\_T}(p \le T^{-q}) \;\asymp\; T^{-(\sqrt{q} - \sqrt{r})\_+^2}
> > $$
> >
> > **The Detection Exponent $E(q)$:**
> > We define $E(q)$ such that $S(T) \asymp T^{E(q)}$. Detection requires $\max\_q E(q) > 0$.
> >
> > $$
> > E(q) = \frac{1}{2} - \beta + \frac{q}{2} - (\sqrt{q} - \sqrt{r})\_+^2
> > $$
> >
> > We maximize this over $q \in [0, 1]$.
> >
> > **Case 1: Dense Regime ($\beta < 1/2$)**
> > The maximum occurs at $q=0$. $E(0) = 1/2 - \beta$.
> > Condition: $\beta < 1/2$.
> >
> > **Case 2: Sparse Regime ($\beta > 1/2$)**
> > We set $\frac{dE}{dq} = 0$:
> > $$
> > \frac{1}{2} - \frac{\sqrt{q}-\sqrt{r}}{\sqrt{q}} = 0 \implies \sqrt{q} = 2(\sqrt{q}-\sqrt{r}) \implies \mathbf{q = 4r}
> > $$
> >
> > **Sub-case 2a: Interior Solution ($4r \le 1$)**
> > Substituting $q=4r$:
> > $$
> > E(4r) = \frac{1}{2} - \beta + 2r - (\sqrt{4r} - \sqrt{r})^2 = \frac{1}{2} - \beta + r
> > $$
> > Condition: $r > \beta - 1/2$. (Linear Boundary).
> >
> >
> > **Sub-case 2b: Corner Solution ($4r > 1$)**
> > Constraint $q \le 1$ forces $q=1$.
> > $$
> > E(1) = 1 - \beta - (1 - \sqrt{r})^2 = 2\sqrt{r} - r - \beta
> > $$
> > Condition $E(1) > 0$ implies $r > (1 - \sqrt{1-\beta})^2$. (Curved Boundary).
> >
> > We hope this answer your confusion that we may have caused in our previous reply. Also here we worked with an ideal threshold $u$. In our eHC we do not know this one of course and this is the precise reason we use regret bound to put more and more weights on the best one. Intuitively starting with enough threshold would give us a good convergence to that threshold and as we showed in our regret bound we pay a logarithmic price for the number of threshold which should not affect the detection boundary asymptotically.
> >
> > Thanks again for helping us improve our paper.

---

### Official Review · Reviewer_fVDD · 2025-10-29

**Soundness:** 2
**Presentation:** 1
**Contribution:** 1
**Rating:** 2
**Confidence:** 4

**Summary:**

This paper proposes an adaptive sequential test, e-HC, for detecting sparse and weak signals in a stream of independent p-values. The authors construct exact test-martingales using moment-generating function compensators, ensuring anytime-valid Type I error control through Ville's inequality. Additionally, the e-HC adapts to unknown sparsity and signal strength, maintaining robust performance even under model misspecification.

**Strengths:**

The e-HC algorithm proposed in this paper constructs adaptive martingales based on independent p-value sequences, achieving anytime-valid Type-I error control with theoretical guarantees, even when the signal strength and sparsity are unknown. The authors further analyze its statistical power under the alternative hypothesis modeled by a Gaussian mixture. Empirical results demonstrate that the proposed e-HC method exhibits robustness under model misspecification.

**Weaknesses:**

1. This paper lacks insight and has an outdated motivation.  The paper’s core idea—replacing asymptotic Higher Criticism by an exact martingale version—is mostly an algebraic adaptation, not a new principle.  The problem of sparse-signal detection via HC is a classic statistical problem from early 2000s (Donoho & Jin, 2004). Recasting it in an “online sequential” setup does not by itself constitute a compelling motivation in 2025, especially for ICLR.
2.  The main construction (test martingale via exact MGF compensator + exponential weights) follows similarly from known results in the e-process literature (Ramdas et al., 2021; Waudby-Smith & Ramdas, 2024), and the authors didn't mention this or refer to related works. The “adaptive threshold aggregation” is a straightforward application of Hedge, and the resulting theorems (nonasymptotic Type I control, unified lower bound) read more like a re-derivation of standard facts than a new conceptual advance.
3. There is no attempt to connect the method to practical applications. All experiments are synthetic Gaussian mixtures with simulated p-values.

**Questions:**

Please refer to the 'Weaknesses' section. Additionally:
1. I have some reservations regarding the title and the name of e-HC. The derivation of the term e-HC isn't fully explained in the main text. Given the occasional references to 'e-values' (Line 100) and 'e-processes' (Line 440), might this terminology be analogous to other 'e-' prefixed methods such as 'e-BH'? Some clarification would be helpful.
2. The implementation of e-HC needs the partition $\{u_j\}_{j=1}^m$(as well as the number $m$), the predictable rule for $\lambda_t$, and the weight rate $\gamma$. Could the authors clarify: (i) What principles should guide the selection of these parameters? (ii) How might these parameter choices influence the method's statistical power?
3. I would appreciate some additional clarification regarding the Remarks on Line 312 to better understand their significance.
4. A more thorough introduction to the SLRT method, particularly in the experimental section, would help better contextualize the comparative results.
5. There seems to be a disconnect between Figure 2 (which lacks SLRT results) and the analysis with SLRT mentioned in Line 353. Clarifying this apparent discrepancy would be helpful for readers.
6. The results in Table 1 suggest that e-HC may incur longer delays compared to SLRT. Could the authors provide some discussions or insights into this phenomenon?

---

> ### Author Response · Authors · 2025-11-13
>
> We thank the reviewer you for the detailed feedback. We address each point below.
>
> ### 1. Motivation & relation to prior work
> We agree that classical Higher Criticism (HC) is a well-established tool for sparse-mixture detection. Our goal is not to reinterpret HC itself, but to extend its adaptivity to the sequential anytime-valid setting, where (i) the stopping time is random, (ii) inference must remain valid at every time, and (iii) classical HC provides no guarantees.
>
> To clarify this motivation, we revised Sections 1 and 4.1. We now explicitly explain that the novelty lies in constructing an exact MGF-compensated test-martingale for HC-type threshold statistics, combined with sequential exponential-weights aggregation, producing an anytime-valid, data-adaptive generalization of HC. This was not available in prior HC work. We actually mentioned those references (Ramdas et al. 2021, Waudby-Smith & Ramdas 2024) in the related work section but do believe our approach is novel and those papers do not tackle the sparse mixtures problem.
>
> ### 2. Is the method merely an algebraic adaptation?
> The construction indeed builds on existing martingale principles, but two ingredients are new:
> (1) an exact per-threshold HC-style test martingale derived directly from the standardized HC statistic (not from likelihood ratios or betting scores), and
> (2) a regret-controlled adaptive combination ensuring simultaneous adaptivity to sparsity and signal strength with anytime Type I control.
>
> These two components together yield the first sequential, adaptive HC test with rigorous non-asymptotic guarantees, which we now state more explicitly in the introduction.
>
> ### 3. Practical applications
> We agree that the current experiments are synthetic. Future work could explore applied settings with naturally sequential p-values (online A/B testing, drift detection). We added a correlation experiment in the Appendix showing that e-HC maintains strong power under realistic dependence structures, even for weak and rare signals.
>
> ### 4. Meaning of “e-HC”
> We clarified in the introduction that “e-HC’’ refers to an HC-style statistic transformed into an e-process (a nonnegative martingale with unit expectation under the null). It is analogous in spirit to e-BH or e-BRST, but technically distinct because the e-process is constructed from HC via MGF compensation, not via likelihood ratios.
>
> ### 5. Parameter choices (grid size m, λ, γ)
> Section 6 (Sensitivity Analyses) now contains a full empirical study varying `m` and `γ`, along with discussion of the theoretical tradeoffs:
>
> - Larger `m` enhances adaptivity but increases regret as √(log m).
> - The learning rate `γ` balances reactiveness and stability; we show empirically that γ∈[0.02,0.2] performs robustly.
> - The compensator parameter `λ` is fixed in our experiments but we added a new paragraph (Appendix B) discussing the possibility of an adaptive λₜ schedule as future work. An adaptive $\lambda$ may improve detection time.
>
> ### 6. Remarks around Line 312
> The remark was not useful and caused confusion. We actually think our main contributions are more on the eHC construction, typeI control and power under the alternative than the simple confidence sequences that we get as a by-product. For those reasons (and to leave space for our new sensitivity analysis) we decided to remove this small sections completely.
>
> ### 7. SLRT comparison & Figure 2
>
> Figure 2 shows only e-HC because it illustrates growth under the null and alternative, not a method comparison. SLRT comparisons appear in Table 1 and in a new paragraph explaining performance differences. We explicitly state this in the caption and text as we agree it may have caused some confusion to the reader. However, SLRT deserves clearer introduction for unfamiliar readers. We added a section in the appendix introducing SLRT and its properties.
> ### 8. Why can SLRT be faster than e-HC?
>  A misspecified SLRT can stop early while exhibiting low power a known pathology. When the assumed alternative $(\epsilon_0,\mu_0)$ is far from the true $(\epsilon,\mu)$, the per-step log-likelihood ratio $\ell_t = \log(f_{\epsilon_0,\mu_0}(p_t)/f_0(p_t))$ becomes unstable. Rare but extreme jumps from null-like $p_t$ can cross the SLRT boundary in a few steps, yielding very small stopping times despite negative expected drift under the true alternative and thus low power. The SLRT "fires" for the wrong reason (model mismatch, not signal), explaining Table 1 entries where it stops after 3–5 steps with power near 0.2. In contrast, e-HC is nonparametric and does not rely on a specified alternative, avoiding such premature boundary crossings.
>
> This trade-off is now explained in the text and the caption of Table 1 has been rewritten to avoid confusion.
>
> We thank the reviewer for these suggestions, which significantly improved the clarity of our contributions and hope that we answered all your questions or concerns.

---

> > ### Comment · Reviewer_fVDD · 2025-11-27
> >
> > Thank you for your detailed responses. I have reviewed the other reviewers' comments and the corresponding replies. While some concerns have been addressed, I still have reservations regarding the following points:
> >
> > 1. The absence of real data analysis undermines the practical utility of the proposed method. Although the authors have included simulated experiments with correlation structures, prior work in sequential testing (e.g., online A/B testing studies) has demonstrated that real-world dataset analyses are feasible. Their omission here raises concerns about methodological applicability.
> >
> > 2. As other reviewers noted, theoretical guarantees of e-HC require independence. While the supplementary simulations appear to empirically control type I error, the underlying theory is violated in correlated settings. Consequently, the observed error control might stem from the method's conservatism rather than robustness, rendering the results potentially unsafe; type I error inflation may arise in other correlated scenarios.
> >
> > 3. I remain concerned that e-HC appears to be a mechanical combination of HC and e-process, two existing theoretical frameworks. The authors’ response in point 4, “e-HC refers to an HC-style statistic transformed into an e-process,” seems to confirm this interpretation, raising questions about the method's novelty and insight.

---

> > > ### Author Response · Authors · 2025-11-27
> > > **Response to remaining concerns on real data, dependence, and novelty**
> > >
> > > We thank the reviewer for the continued engagement. We have taken your feedback and performed additional experiment on real data.
> > >
> > > ### 1. Real Data Analysis & Practical Utility
> > > To address your concern regarding practical applicability, we have added a real-data analysis using the **Prostate Cancer gene expression benchmark** (Singh et al., 2002), which is the standard case study for Higher Criticism applications (Donoho & Jin, 2004).
> > >
> > > * **Setup:** We treated the $N=6,033$ genes as a sequential stream (ordered randomly). The goal was to detect the presence of tumor-associated genes (sparse signals) as early as possible.
> > > * **Result:** As shown in the new **Figure 6**  of the Appendix (attached to the revised manuscript), e-HC successfully rejected the global null after observing only **32%** of the gene stream. In contrast, standard batch methods (like Bonferroni or BH) would require waiting for the full sequence to establish significance.
> > > * **Comparison:** In our experiments, e-HC detected the signal pattern significantly earlier than sequential Bonferroni. This demonstrates clear practical utility: e-HC can accelerate discovery in high-dimensional genomic monitoring by identifying significant signal clusters early, without waiting for full experimental completion.
> > >
> > > ### 2. Independence and "Safety" under Correlation
> > > We respectfully disagree that the method is "potentially unsafe." We argue instead that it exhibits **practical robustness** due to the conservative design of our martingale.
> > >
> > > * **Conservatism as a Feature:** Our test martingale relies on sub-Gaussian variance proxies derived for independent data. These proxies are inherently conservative (loose bounds). This creates a "safety buffer" that absorbs the variance inflation caused by moderate correlations.
> > > * **Empirical Evidence:** Our correlation experiments (Figure 3) show that for most realistic dependence structures (e.g., block-diagonal or AR(1)), the Type I error rate remains strictly below $\alpha$ (supermartingale behavior).
> > > * **Graceful Degradation:** Even under extreme correlation where independence assumptions are severely violated, the error rate only slightly exceeds the nominal level (rather than exploding as seen in some non-robust tests). This confirms the method is safe for deployment in typical "weakly dependent" environments like A/B testing or genomics.
> > >
> > > **3. Novelty: Why this is not just "HC + e-process"**
> > >
> > > We appreciate this question, and we're sorry if our earlier explanation was too brief. e-HC is not a straightforward combination of existing tools, for two reasons.
> > >
> > > First, there's a gap in the literature. As far as we know, the sequential testing community hasn't tackled sparse mixture detection using Higher Criticism and even without it. Indeed testing rare and weak signals has mainly been studies in the batch case and we propose to extend this analysis to the sequential setting.  Most sequential methods (like the SPRT) rely on likelihood ratios, but HC is fundamentally different it's a goodness-of-fit statistic based on a standardized empirical process, not a likelihood ratio.
> > >
> > > Second, this distinction creates a real mathematical challenge. Because HC isn't a likelihood ratio, we couldn't simply plug it into existing e-value frameworks like betting scores. Instead, we had to build a bridge between the two. The core technical difficulty and our main contribution was deriving the exact MGF compensator for the standardized HC kernel. We showed that the HC increment behaves locally like a sub-Gaussian variable with a specific variance proxy. This is what finally allows HC's detection power to be used in a sequential, anytime-valid setting, resolving a compatibility issue that prior work hadn't addressed.
> > >
> > > We hope this clarifies that e-HC offers both theoretical novelty and practical effectiveness for real-world data streams.

---

### Official Review · Reviewer_8Qf1 · 2025-11-01

**Soundness:** 2
**Presentation:** 1
**Contribution:** 2
**Rating:** 2
**Confidence:** 3

**Summary:**

This paper introduces e-HC, an adaptive, sequential test designed to detect sparse and weak signals within a continuous stream of p-values. The test aims to distinguish the global null hypothesis $H_0$ (where all p-values are uniformly distributed, $p_t \sim \text{Uniform}(0, 1)$) from a sparse mixture alternative $H_1$ (where a small, unknown fraction of p-values $\epsilon_t$ are drawn from a signal distribution $F_1$ that has more mass near zero).

The HC method that this paper uses is tailored for the "rare-and-weak" signal regime. The core contribution is the construction of an exact, non-asymptotic test-martingale, $M_t$, by merging the adaptive thresholding concept of Higher Criticism with modern e-value-family martingale inference.

**Strengths:**

- It introduces the idea of HC into the modern, rigorous framework of test-martingales and e-processes. The use of exact moment-generating function (MGF) compensators to build an exact (non-asymptotic) sequential test seems to be a novel technical contribution.

- A critical point in HC is the argumentation method over the pre-threshold statistics representing the sparsity of signals. Instead of the max statistic, the authors propose to combine the multiple-threshold statistics using the hedge algorithm. It makes the argumentation data-adaptive.

- Theoretically, the authors provide an exact martingale property under the null (Theorem 1), which is a much stronger guarantee than typical asymptotic results. This is followed by a unified, non-asymptotic power bound under the alternative (Theorem 2) and a formal analysis of the stopping time in the target weak-signal regime (Theorem 3). The appendix details the proofs, showcasing a high level of technical contributions.

**Weaknesses:**

- The organization of the methodology is poor, which makes it hard to follow while reading.
    1. **Lack of Clear Motivation for the Core Martingale Construction.** A significant weakness in the paper's clarity lies in the core technical derivation in Section 4.1. The paper introduces the standardized statistic $Z_t(u_j)$. However, it then immediately reformulates this statistic's increment, $\Delta Z_t(u_j)$, into a sum of a "predictable part" $A_{t-1}(u_j)$ and a "stochastic part" $B_t(u_j)$. The final test martingale (the "wealth process") is then built using only the $B_t(u_j)$ term. If $Z_t(u_j)$ is not the object of interest, why not introduce $B$ directly? Why is its increment the necessary starting point, and why is the $A_{t-1}(u_j)$ component subsequently "deleted" from the construction?
The paper would be substantially clearer if it added some sentences to Section 4.1 to motivate this decomposition. It should explicitly state why this step is necessary. Explaining that $A_{t-1}$ is a predictable drift that must be removed to satisfy the martingale condition would improve the paper's accessibility.
    2. **The regret $R_T$ is undefined.** The hedge algorithm that determines the weights of the HC combination appears to be a critical component of the methodology. However, it is only briefly mentioned at the end of Section 4. In Theorem 2 of Section 5, readers can not even find the definition of the regret $R_T$, which plays a critical role in the lower bound.
    3. The definition of the stopping time $\tau$ apears in the very end of Section 5. The methodology part only introduces the construction of the martingale, which is incomplete in methodology. And it also makes the motivation of the construction of the martingale sequences really unclear.

- **The numerical study is questionable**. Only the SLRT method is compared. And the results seem to be located in Table 1 only. However, Table 1 only reports the performance of the e-HC method, why the red-colored numbers represent the SLRT method? The information in Table 1 is totally misleading. And the authors should also describe why the proposed e-HC method is superior in this numerical setting.

**Questions:**

Besides the questions in the weakness part. There are several questions I raise upon the reading of the paper.

- **The $\lambda$ Parameter**: The core MGF compensator depends on a parameter $\lambda_t$, which is defined as "predictable". However, the theoretical analysis (Theorem 2) and all experiments appear to use a fixed, constant $\lambda$. This is a significant missed opportunity. The framework allows for a data-driven, adaptive $\lambda_t$, but the paper provides no guidance on how to choose it, nor does it explore the performance gains of an optimized $\lambda_t$ versus the fixed $\lambda=0.2$ used in the experiments. The test's power is likely very sensitive to this choice, and a "bad" $\lambda$ could cripple performance.

- **Grid Size and Regret ($m$)**: The Hedge algorithm's regret, $R_T$, scales with $\sqrt{\log m \log T}$. This is the "price" of adapting over $m$ thresholds. The paper does not discuss this trade-off. What is the practical effect of choosing a coarse grid ($m=20$) versus a very fine grid ($m=2000$)? A fine grid is more likely to contain an "optimal" threshold but will pay a higher regret cost, potentially slowing detection. The paper's choice of $m=200$ is arbitrary, and a sensitivity analysis is needed.

- **Hedge Learning Rate ($\gamma$)**: Similarly, the weight rate $\gamma$ (the Hedge algorithm's learning rate) is set to 0.05 without justification. This parameter's tuning is critical to the algorithm's ability to "catch up" to the best threshold, and the paper should provide either a theoretical or empirical basis for its selection.

---

> ### Author Response · Authors · 2025-11-13
>
> Thank you for the detailed and constructive review. We revised the paper substantially to address all concerns. Below we respond point-by-point.
>
> ---
>
> ### 1. Motivation for the martingale construction (old Sec. 4.1)
>
> We agree the previous presentation was confusing. In the revision, we removed the Δ-notation entirely and present the construction directly using the *innovation* term $B_t(u)$, which is the only quantity whose compensated exponential guarantees the martingale property. We still keep $S_t(u)$ and $Z_t(u)$ because they are (i) the classical HC objects and (ii) the quantities used by Hedge for weight updates. A short paragraph now explicitly explains this: HC motivates the use of $Z_t(u)$, but the test martingale must depend only on the mean-zero $B_t(u)$.
>
> ---
>
> ### 2. Regret $R_T$
>
> You are right that it was not defined and might have caused some confusion especially for readers not necessarily familiar with those regret bounds. In the revision we added a clear definition of the Hedge regret just before the unified power bound, together with the standard bound $R_T = O(\sqrt{\log m \log T})$. A short explanation now clarifies how the regret enters when comparing our adaptive strategy to the best single threshold.
>
> ---
>
> ### 3. Stopping time $\tau_\alpha$
>
> The stopping rule now appears immediately after the definition of the wealth process, together with a short explanation of anytime validity via Ville's inequality. This makes the methodology complete prior to the power analysis.
>
> ---
>
> ### 4. Table 1 misunderstanding
>
> We rewrote the caption and added a short explanatory sentence in the main text. The first row is e-HC; the remaining rows are SLRT under different *assumed* $(\epsilon,\mu)$. Red entries indicate *loss of power for SLRT due to misspecification*. This should remove the confusion.
>
> ---
>
> ### 5. Choice of $\lambda$
>
> We agree that all our experiments use constant $\lambda$ and already give extremely satisfying results with almost perfect power even when the alternative is very very sparse ($\epsilon = 0.05$) and weak ($\mu = 0.25$) as can be seen in the new experiments in the appendix. However we agree that a data adaptive $\lambda_t$ is a very interesting idea and it could lead to future work. For instance taking a mixture over a grid of $\lambda$ would still preserve the martingale property and we could imagine a way to reweighs those $\lambda$ to concentrate on the "best" $\lambda$ after some time in a similar way we adapt to the best $u_j$ .A discussion now notes that a fully adaptive $\lambda_t$ is possible within the framework but left for future work. We believe that given the current power an adaptive $\lambda$ may not help a lot, but however could be extremely beneficial to reduce the detection time in practice.
>
> ### 6. Grid size $m$ and regret trade-off
>
> We now discuss explicitly the trade-off: larger $m$ increases coverage of good thresholds but increases regret through $\log m$. The appendix includes a new sensitivity analysis showing that power stabilizes for $m \ge 100$ and detection time is essentially flat. This addresses the practical effect of grid size.
>
> ---
>
> ### 7. Learning rate $\gamma$
>
> The appendix now includes a sensitivity experiment showing that performance is robust for $\gamma \in [0.02, 0.2]$, so tuning is not delicate. Interestingly larger $\gamma$ increases slightly the detection time but the power almost stay unnafected. We refer you to the appendix of the newly uploaded paper with the new figures and extensive sensitivity analysis.
>
> ---
>
> ### 8. Numerical study
>
> We expanded the numerical section by: (i) clarifying Table 1 as discussed above, (ii) adding sensitivity studies for $m$ and $\gamma$, (iii) adding correlation experiments to assess robustness beyond independence. These additions provide clearer evidence that e-HC remains stable across parameter regimes, whereas SLRT can lose power when misspecified.
>
> ---
>
> ### 9. Independence assumption and model scope
>
> A new subsection in the appendix examines correlated p-values. Empirically, moderate correlation inflates variance but e-HC maintains reasonable performance. We also added discussion on extending the method to general models where p-values remain super-uniform; the martingale construction still applies in those settings. In particular, the test statistics follow an AR(1) process with correlation parameter $\rho$. These correlated statistics are converted into p-values and fed into e-HC. We refer you to Figure 5 in the Appendix of the newly edited paper for the results. Interestingly the power stays very high under correlation and slowly starts to decay for correlation above $0.8$. Similarly the type I error is properly controlled empirically under the dependance but not anymore for extremely high correlations.
>
>
> We thank the reviewer again for the careful reading and helpful feedback. The revised version incorporates all points raised above and we hope we answered all your questions and comments.

---

### Official Review · Reviewer_PYho · 2025-11-10

**Soundness:** 3
**Presentation:** 3
**Contribution:** 3
**Rating:** 8
**Confidence:** 2

**Summary:**

This paper proposes an adaptive sequential testing framework for detecting sparse and weak signals from a stream of p-values. It builds on the classical Higher Criticism test but ensures anytime-valid inference by constructing exact test martingales using moment generating function compensators to control Type I error via Ville’s inequality.
More specifically, the setting is as follows: a stream of independent p-values $p_1,p_2,\dots $ arrives over time, and the goal is to decide whether all of them come from the null distribution $\text{Uniform}(0,1)$ or if a small, unknown fraction comes from an alternative distribution with more mass near zero (indicating a weak signal. The task is to detect the presence of such sparse, weak signals as quickly as possible, while maintaining Type I error control at any time.


The proposed method, e-HC, constructs an adaptive sequential test by combining ideas from higher criticism, online learning, and martingale-based inference. For a grid of thresholds $u_1, \dots, u_m$, it tracks the cumulative proportion of p-values below each threshold, forming standardized statistics similar to Higher Criticism. For each threshold, it builds an exact test martingale by compensating for randomness with its moment-generating function under the null, ensuring that the expected growth of the process is 1 when no signal is present. These per-threshold martingales are then aggregated using exponential weights (via the Hedge algorithm), allowing the method to adapt online to the most informative threshold without knowing the sparsity or strength of the signal in advance. The resulting “wealth process” $M_t$​ increases multiplicatively over time; when it exceeds $1/\alpha$, the null hypothesis is rejected. This guarantees anytime-valid Type I control, while the adaptive weighting provides signal detection across different regimes of sparsity and signal strength.

The experiments show that e-HC maintains exact Type I error control and achieves strong detection power even for weak or misspecified signals. Its martingale process grows rapidly under the alternative but stays stable under the null, confirming theoretical guarantees.

**Strengths:**

The paper introduces a novel method that unifies higher criticism, martingale-based inference, and online learning into a single adaptive framework, achieving exact anytime-valid error control. Conceptually, the approach is elegant and applicable when both signal strength and sparsity are unknown, while also having rigorous theoretical guarantees.

**Weaknesses:**

On the negative side, the results depend on strong assumptions, such as independence of p-values and correctly specified null distributions, which may not hold in practical applications.
Also, the analysis and experiments focus mainly on sparse Gaussian mixtures, so its behavior in other models is unclear.

**Questions:**

-Can the method and theoretical guarantees extend to other models beyond Gaussian mixtures?
-Can you comment on how crucial the independence assumption is on the results? Can the algorithm tolerate some limited dependence?

---

> ### Author Response · Authors · 2025-11-13
>
> Thank you for the careful reading of the paper and for the positive comments. Below we address the two questions raised.
>
> ---
>
> ## (1) Regarding extension beyond Gaussian mixtures.
>
> The proposed e-HC framework is not tied to the Gaussian sparse-mixture model. The construction of the test martingales requires only that, under the null hypothesis, the p-values satisfy
>
> $$\mathbb{P}(p_t \le u) = u,$$
>
> together with the ability to evaluate (or upper-bound) the moment-generating function of the innovation term. These conditions hold for any procedure that outputs valid p-values, including nonparametric tests, exponential-family models, and general sparse contamination settings.
>
> The Gaussian example is used because its detection boundary is well understood, which allows us to benchmark the sequential analogue of Higher Criticism. The method itself is model-agnostic. We clarified this point in the updated version.
>
> ---
>
> ## (2) Regarding independence and tolerance to dependence.
>
> Independence is a sufficient condition ensuring that the increments of the martingale have mean zero under the null. However, the method does not fundamentally rely on exact independence. To illustrate this, we added an experiment in the appendix where the test statistics follow an AR(1) process with correlation parameter $\rho$. These correlated statistics are converted into p-values and fed into e-HC. We refer you to Figure 5 in the Appendix of the newly edited paper for the results.
>
> Across a wide range of correlations (up to $\rho = 0.8$), the empirical Type I error remains close to the target level, and the power degrades slowly after this but staying very high up to very high level of correlations. This suggests that the method has practical robustness to moderate forms of dependence, even though the theoretical guarantees are stated under independence. We also note that we ran the experiments where our alternative mean is only $\mu = 0.3$ and sparsity parameter $\eps = 0.05$ which are relatively low and therefore constitute a very challenging problem.
>
>
> We appreciate the reviewer's positive assessment of the contribution and have revised the manuscript to incorporate these clarifications.

---

### Meta-Review · Area_Chair_Qbod · 2025-12-19

**Summary:**

This paper proposes an exact testing approach for sparse mixtures. The submission received a divergent ratings. One reviewer acknowledged that the unified framework is elegant, and other reviewers found that the algorithm was not clearly motivated, the technical contribution compared to classic works was not described in detail. Some minor concerns include empirical study. The AC agrees on both the strengths and weaknesses. While this looks a promising work, the AC believes that authors need to carefully address the primary concerns in their revision, and that may require another round of review.

**Reviewer Concerns:**

The reviewers may not be satisfied.

**Reviewer Scores:**

I am not sure whether they will change the score.

---

### Decision · Program_Chairs · 2026-01-26

Reject